# Optimizing spca-based continual learning: a theoretical approach

**Chunchun Yang**
University of Science and Technology of China
Huawei Noah's Ark Lab
yangchunchun4@huawei.com

**Malik Tiomoko**
Huawei Noah's Ark Lab
Paris, France
malik.tiomoko@huawei.com

**Zengfu Wang**
University of Science and Technology of China
Hefei Institutes of Physical Science, Chinese Academy of Sciences, China
zfwang@ustc.edu.cn

## Abstract

*Catastrophic forgetting* and the *stability-plasticity* dilemma are two major obstacles to continual learning. In this paper, we first propose a theoretical analysis of a SPCA-based continual learning algorithm using high-dimensional statistics. Second, we design *OSCL* (Optimized Spca-based Continual Learning) which builds on a flexible task optimization based on the theory. By optimizing a single task, *catastrophic forgetting* can be prevented theoretically. While optimizing multi-tasks, the trade-off between integrating knowledge from the new task and retaining previous knowledge of the old tasks can be achieved by assigning appropriate weights to corresponding tasks in compliance with the objectives. Experimental results confirm that the various theoretical conclusions are robust to a wide range of data distributions. Besides, several applications on synthetic and real data show that the proposed method while being computationally efficient, achieves comparable results with some state of the art.

## 1 Introduction

**Continual learning paradigm.** Machine learning methods generally learn from samples of data randomly drawn from a stationary distribution. However, this scenario is rare in reality. Continual learning (CL) is a particular machine learning paradigm in which data continuously arrive in a possibly non *i.i.d.* way and knowledge is accumulated over time (Schlimmer & Fisher, 1986; Ebrahimi et al., 2019; Lee et al., 2020; De Lange et al., 2021). For designing real-world machine learning systems that mimic humans, continual learning is essential. On the one hand, humans continue to acquire knowledge and solve new problems throughout their lifetimes. The goal of continual learning is to mimic the capacity of humans to learn from a non-stationary data stream without forgetting catastrophically the learned knowledge (Titsias et al., 2019; Lee et al., 2020). On the other hand, when deploying a trained model in real applications, the distribution of data will consistently drift over time. Therefore, the machine learning algorithm must be able to adapt continuously to these changes (Kirkpatrick et al., 2017; Lesort et al., 2020).

**Challenges in continual learning.** One of the major challenges of continual learning is to avoid *catastrophic forgetting* (McCloskey & Cohen, 1989; Chen & Liu, 2018; Aljundi, 2019). This occurs when the performance of previous tasks is severely degraded during the learning process. To take into account both the current task and the previous tasks, the *stability-plasticity* dilemma was introduced (Nguyen et al., 2017; Rajasegaran et al., 2019). More specifically, *plasticity* refers to the ability of integrating new knowledge, and *stability* to the capacity of retaining previous knowledge (which is related to *catastrophic forgetting*). Note that the term *catastrophic forgetting*, although strongly referenced in the literature in deep neural network models, is a fairly general concept that can occur in any machine learning algorithm as it has been noted in shadow single-layer models, such as self-organizing feature maps (Richardson & Thomas, 2008; Chen & Liu, 2018).

**State of the art.** Most works on CL have focused on purely supervised tasks which is the focus of the present work (Kirkpatrick et al., 2017; Nguyen et al., 2017; Lee et al., 2020). Supervised CL can be classified into three main categories based on how knowledge and data are updated and stored: replay, regularization, and dynamic architecture-based methods (De Lange et al., 2021). Replay-based approaches address the *catastrophic forgetting* by saving and reusing the previously seen data while learning a new task (Lopez-Paz & Ranzato, 2017; Isele & Cosgun, 2018; Titsias et al., 2019). Regularization-based methods penalize the update of crucial weights to alleviate forgetting by introducing an extra regularization term in the loss function (Kirkpatrick et al., 2017; Ebrahimi et al., 2019). Dynamic architecture-based methods flexibly update the learning model as new tasks are added based on the task complexity and the relation between the tasks (Rajasegaran et al., 2019; Lee et al., 2020). Although successful cases of supervised continual learning were reported in articles, the algorithms tend to feature unpredictable behavior, generally requiring a host of additional inconvenient hyperparameters and requiring a high computational cost. More importantly, supervised continual learning methods lack theoretical guarantees of avoiding *catastrophic forgetting*.

**Contributions of the paper.** In this paper, we introduce a novel CL method based on Supervised Principal Component Analysis (SPCA). We provide a theoretical analysis of the proposed method using high-dimensional statistics. This analysis allows us to predict in advance the performance of the algorithm. Furthermore, we develop a label optimization scheme based on the theory that avoids *catastrophic forgetting* theoretically. As a result, we obtain a simple and efficient continual learning algorithm free of hyperparameters, at a low computational cost named *OSCL*. Moreover, the theory allows us to weight the tasks during the learning process according to the user's preferences and priorities. This is in line with the resolution of the *stability-plasticity* dilemma. As such, the main contributions of the paper can be summarized as follows. We propose a simple continual learning algorithm, computationally inexpensive based on SPCA, and provide a theoretical analysis (to obtain exact classification error rather than bounds) using high dimensional statistics. Using the theoretical results, we develop a label optimization scheme that provably prevent *catastrophic forgetting* and also allows for a weighting of the tasks during learning. Several applications are presented to corroborate the practical usefulness of the approach in terms of efficiency and flexibility.

**Outlines.** The remainder of the paper is organized as follows. Section 2 discusses several works in the continual learning literature and highlights the differences and contributions of the present paper. Section 3 introduces and formalizes the continual learning framework, and furthermore proposes a simple continual learning algorithm based on SPCA. Section 4 proposes a theoretical analysis of this algorithm as well as flexible optimization tools to benefit from all tasks. Section 5 provides several applications to corroborate the different conclusions of the paper.

**Notations.** Matrices will be represented in bold capital letters (*e.g.,* matrix $\boldsymbol{A}$). Vectors will be represented in bold lowercase letters (*e.g.,* vector $\boldsymbol{v}$) and scalars will be represented without bold letters (*e.g.,* variable $a$). The *canonical vector* of size $n$ is denoted by $\boldsymbol{e}_m^{[n]} \in \mathbb{R}^n$, $1 \leq m \leq n$, where the $i$-th element is 1 if $i = m$, and 0 otherwise. The diagonal matrix with diagonal $\boldsymbol{x}$ and 0 elsewhere is denoted by $\mathcal{D}_{\boldsymbol{x}}$. Generally, the subscript $t$ refers to the task number, and the superscript $j$ to the class index. As an example, $\boldsymbol{x}_{t\ell}^j$ denotes the $\ell$-th sample of class $j$ for task $t$. $\mathbb{1}_n \in \mathbb{R}^n$ is the vector of all ones and the matrix $\boldsymbol{\Sigma}_{\boldsymbol{v}} \in \mathbb{R}^{n \times n}$ denotes the covariance matrix of the random vector $\boldsymbol{v}$. $[n]$ denotes the set $\{1, \ldots, n\}$. $n_t$ denotes the number of samples in task $t$, $n_{tj}$ is the number of samples in class $j$ of task $t$.

## 2 RELATED WORKS

### 2.1 CONTINUAL LEARNING

As mentioned in the introduction, we focus on the state of the art of supervised CL, which is divided into three main concepts that we develop in detail in this section. The appendix provides a more detailed picture for interested readers.

**Replay-based methods.** Replay-based approaches aim to retain a certain amount of historical examples, extracted features, or generated examples to reduce forgetting when training the model with new data. Three challenges need to be solved. The first problem is selecting appropriate previous samples. In this context, Isele & Cosgun (2018) proposed four strategies for choosing which data will be stored. Aljundi et al. (2019) proposed sample selection as a constraint reduction problem.

The second challenge is to find a suitable format to store data from past tasks. In this direction, Shin et al. (2017); Ayub & Wagner (2020) used a deep generative model to sample pseudo-data points to replay the past experience while Farajtabar et al. (2020) stored a set of gradients of each task to minimize *catastrophic forgetting*. The last challenge is to find a better way to use historical data. Lopez-Paz & Ranzato (2017); Chaudhry et al. (2018); Farajtabar et al. (2020) avoided interference between tasks by projecting the new gradients in the feasible region constrained by the gradient(s) of previous tasks. Wang et al. (2022) achieved a better stability-plasticity trade-off using an attention-based framework while Hersche et al. (2022) composed a frozen feature extractor, a trainable fixed size fully connected layer, and a re-writable dynamically growing EM (Explicit Memory) that stores as many vectors as the number of classes encountered. It is important to note that our method does not use any samples from previous tasks and only stores statistical data. Therefore our method is different from the aforementioned algorithms.

**Regularization-based methods.** Regularization-based approaches generally impose a regularization constraint on the objective function to maintain a balance between *stability* and *plasticity*. The literature focuses on two main issues. In the first place, it is necessary to evaluate the importance of parameters in the previous model and, in the second place, to limit the adjustment of critical weights. With Kirkpatrick et al. (2017), such importance is computed from the diagonal of the Fisher information matrix and important weights are constrained to change slowly while learning new tasks. Nguyen et al. (2017) extended online variational inference to handle more general continual learning tasks. Ebrahimi et al. (2019); Zeno et al. (2018) used uncertainty as the importance of the weights and subsequently, adapted the learning rate whereas Jung et al. (2020) considers the average activation value as the importance of a node. Our algorithm does not use any regularization and therefore differs from regularization-based approaches.

**Dynamic architecture-based methods.** Dynamic architecture-based approaches adjust the model architecture according to the complexity of the tasks, and the relation between them. Two questions are critical in this approach: the timing to adjust the model architecture and the procedure to do so. In Fernando et al. (2017) the changing procedure uses evolutionary strategies to select pathways through the network and freeze older pathways. In contrast, Yoon et al. (2018) and Lee et al. (2019) don't modify systematically the architecture given new tasks but take into account the difficulty of the tasks and the relation between them. More specifically, Yoon et al. (2018) used group sparse regularization to dynamically expand the architecture when the accumulated knowledge alone cannot sufficiently explain the new task. Similarly, Lee et al. (2019) expands the number of experts under a Bayesian non-parametric framework, which determines the complexity of the model based on the data. Our method keeps the same model from one task to another and does not fit into approaches based on dynamic architectures.

Although the idea of label optimization has been introduced in a transfer learning framework (Tiomoko et al., 2020), this paper is the first to propose a line of label optimization in a CL context. As a result, our study is not directly part of the aforementioned CL classification methods but can be considered as a new research line for CL.

## 2.2 SUPERVISED PRINCIPAL COMPONENT ANALYSIS

Our continual learning approach is based on SPCA. Barshan et al. (2011a) is one of the first to propose SPCA, which is a generalization of principal component analysis (PCA) and aims at finding the principal components with maximum dependence on the response variables. Ritchie et al. (2019) presented a manifold optimization approach to SPCA that simultaneously solves the prediction and dimension reduction problems, which is general enough for both regression and classification settings. For a more detailed review, we refer to Ghojogh & Crowley (2019) which explains PCA, SPCA, kernel PCA, and kernel SPCA. The advantage of SPCA in a CL context is its ease of transferring knowledge from previous tasks to the next in addition to its simplicity of theoretical analysis, which is a major advantage over more complicated algorithms such as SVM, LDA, logistic regression, or neural networks.

## 2.3 ON LARGE DIMENSIONAL ANALYSIS OF MACHINE LEARNING ALGORITHMS

The field of theoretical analysis of machine learning algorithms has become increasingly large with a significant impact on the understanding and optimization of algorithms. Different fields of research

are explored, among which we find approaches using the theory of random matrix (Mai & Couillet, 2021; Couillet & Benaych-Georges, 2016; Liao & Couillet, 2019; Seddik et al., 2020; Tiomoko et al., 2020; Niyazi et al., 2021), approaches based on physical statistics (Cocco et al., 2018; Zdeborová & Krzakala, 2016; Malzahn & Opper, 2005; Agliari et al., 2020; Carleo et al., 2019; Agliari et al., 2020; Taheri et al., 2021), and other approaches (Convex Gaussian Min Max theorem notably (Thrampoulidis et al., 2014; Deng et al., 2020)). These approaches generally have the same goal: to predict in advance the classification error or to predict the mean square error in a regression framework to anticipate the behavior of the algorithm and improve it. Our study is part of this wave of asymptotic analyses and is more specifically related to the approaches of the theory of random matrices.

Furthermore, using Bayesian and information theoretic arguments, Pentina & Lampert (2014; 2015); Doan et al. (2021); Alquier et al. (2017); Denevi et al. (2019a;b) established bounds for the classification error in a continual learning framework. Even though these works provide loose bounds and orders of magnitude, they do not provide a satisfying and accurate evaluation of performance, despite their convenience for deciding whether an objective is possible. In this paper, we provide a *precise* and *exact* theoretical classification error for SPCA-based continual learning.

## 3 PRELIMINARIES

### 3.1 CONTINUAL LEARNING FRAMEWORK

Let's consider a $m$-class classification problem in a $d$-dimensional space with training data of task $t$ denoted $\boldsymbol{X}_t = [\boldsymbol{X}_t^1, \ldots, \boldsymbol{X}_t^m] \in \mathbb{R}^{d \times n_t}$ where $\boldsymbol{X}_t^j = [\boldsymbol{x}_{t1}^j, \ldots, \boldsymbol{x}_{tn_{tj}}^j] \in \mathbb{R}^{d \times n_{tj}}$ are the $n_{tj}$ vectors of class $j \in \{1, \ldots, m\}$. To each $\boldsymbol{x}_{t\ell}^j \in \mathbb{R}^d$ is attached a corresponding "label" (or score) $y_{t\ell}^j \in \mathbb{R}$. We denote in short $\boldsymbol{y}_t = [y_{t1}^1, \ldots, y_{tn_t}^m]^\mathsf{T} \in \mathbb{R}^{n_t}$ the vector of all labels of task $t$.

In a continual learning setting, when dealing with task $t$, the user only has access to the data of task $t$, *i.e.,* $\boldsymbol{X}_t$ and the associated label $\boldsymbol{y}_t$, and needs to predict label $\hat{y}$ for a test data $\boldsymbol{x} \in \mathbb{R}^d$ coming from an already seen task $\tau \in [t]$. If the task identity of the test data is known a priori, we refer the setting to the *Task Incremental Learning (TIL)*. The unknown task label during the inference scenario is divided into the *Domain Incremental Learning (DIL)* where all tasks share the same label space and *Class Incremental Learning (CIL)* when it changes. van de Ven & Tolias (2019) details more deeply the relationships between the different settings.

### 3.2 SPCA-BASED CONTINUAL LEARNING ALGORITHM

For simplicity, we will limit ourselves to the binary case for now. Multi-class learning is handled by using binary-based approaches such as *one-versus-one* and *one-versus-all* which are explained in Section 4 and fully discussed in Appendix C. SPCA (Barshan et al., 2011b) extracts the principal components of the data that are most dependent on the target variable.

More specifically, in a binary classification context, let the data matrix $\boldsymbol{X} = [\boldsymbol{X}_1, \ldots, \boldsymbol{X}_t] \in \mathbb{R}^{d \times n}$, with $n = \sum_{\tau=1}^t n_\tau$, be the collection of the training data of the $t$ tasks already seen and their associated labels or scores $\boldsymbol{y} = [\boldsymbol{y}_1^\mathsf{T}, \ldots, \boldsymbol{y}_t^\mathsf{T}]^\mathsf{T} \in \mathbb{R}^n$, the decision function of SPCA is given for a new test data $\boldsymbol{x}$ by

$$g(\boldsymbol{x}) = \frac{1}{n}\boldsymbol{y}^\mathsf{T}\boldsymbol{X}^\mathsf{T}\boldsymbol{x} = \frac{1}{n}\sum_{\tau=1}^t \boldsymbol{y}_\tau^\mathsf{T}\boldsymbol{X}_\tau^\mathsf{T}\boldsymbol{x}. \tag{1}$$

Since $\boldsymbol{x}_{t1}^{(j)}, \ldots, \boldsymbol{x}_{tn_t}^{(j)}$ are *i.i.d.* data vectors, we impose equal scores $y_{t1}^{(j)} = \ldots = y_{tn_{tj}}^{(j)}$ denoted $\tilde{y}_{tj}$ within every class $j$ of task $t$. As such, we may reduce the score vector $\boldsymbol{y} \in \mathbb{R}^n$ under the form

$$\boldsymbol{y} = \left[\tilde{y}_1^1 \mathbb{1}_{n_{11}}^\mathsf{T}, \ldots, \tilde{y}_t^2 \mathbb{1}_{n_{t2}}^\mathsf{T}\right]^\mathsf{T}$$

for $\tilde{\boldsymbol{y}} = [\tilde{y}_1^1, \ldots, \tilde{y}_t^2]^\mathsf{T} \in \mathbb{R}^{2t}$. This consists in assigning to each class of each task a unique and common label. In a classical way, one class of all tasks will be assigned a score of $-1$, while the other class will be given a score of $1$, *i.e.,* $\tilde{\boldsymbol{y}} = [-1, 1, -1, 1, \ldots, -1, 1]^\mathsf{T}$. The class allocation is therefore performed based on the sign of the decision function $g(\boldsymbol{x})$. We will see that the classical

choice of labels for SPCA based continual learning leads to *catastrophic forgetting* and does not allow us to solve the *stability-plasticity* dilemma. For this reason, we leave a free choice for the labels or scores $\tilde{\boldsymbol{y}}$ and we will optimize them using the theory similarly as done in Tiomoko et al. (2020). This optimization of labels can be intuitively seen as a weight of different classes with respect to their importance in the knowledge preservation.

This being said, Equation 1 can then be rewritten in a more convenient manner as

$$g(\boldsymbol{x}) = \sum_{\tau=1}^{t} \sum_{j=1}^{2} \frac{n_{\tau j}}{n} \tilde{y}_\tau^j \hat{\boldsymbol{\mu}}_\tau^{j\mathsf{T}} \boldsymbol{x}, \qquad \hat{\boldsymbol{\mu}}_\tau^j = \frac{1}{n_{\tau j}} \boldsymbol{X}_\tau^j \mathbb{1}_{n_{\tau j}}. \tag{2}$$

Therefore, when dealing with task $t$, the user just needs to have access to the empirical means $\hat{\boldsymbol{\mu}}_\tau^j$ and the label $\tilde{y}_\tau^j$ for $\tau \leq t$, which allows an important memory saving compared to sending the raw data.

CL has the important challenge of building a flexible optimization method that solves the *catastrophic forgetting* and the *stability-plasticity* dilemma. To achieve this goal, we propose a theoretical analysis of the SPCA-based continual learning algorithm. Precisely, we will characterize the theoretical classification error of each task such that a flexible optimization scheme can be performed based on the theoretical analysis.

To that end, we will make use of the following assumption on the data distribution.

**Assumption 1 (Concentration of $\boldsymbol{X}$)** *For class $\mathcal{C}_j$, $j \in \{1, 2\}$ of task $t$, we assume that all vector $\boldsymbol{x}_{t1}^j, \ldots, \boldsymbol{x}_{tn_{tj}}^j \in \mathcal{C}_j$ are i.i.d. such that $\mathrm{Cov}(\boldsymbol{x}_{t1}^j) = \boldsymbol{\Sigma}_t^j$ and $\mathbb{E}[\boldsymbol{x}_{t1}^j] = \boldsymbol{\mu}_t^j$. Moreover, we assume that there exist two constants $C, c > 0$ (independent of $n, d$) such that, for any $1$-Lipschitz function $f : \mathbb{R}^d \to \mathbb{R}$,*

$$\mathbb{P}_{\boldsymbol{x} \sim \mathcal{D}(\boldsymbol{X})} \left( |f(\boldsymbol{x}) - m_{f(\boldsymbol{x})}| \geq t \right) \leq C e^{-(t/c)^2} \qquad \forall t > 0,$$

*where $m_Z$ is a median of the random variable $Z$.*

Assumption 1 notably encompasses the following scenarios: the columns of $\boldsymbol{X}$ are (a) independent Gaussian random vectors with identity covariance, (b) independent random vectors uniformly distributed on the $\mathbb{R}^d$ sphere of radius $\sqrt{d}$, and, most importantly, (c) any Lipschitz continuous transformation thereof, such as GAN as it has been recently theoretically shown in Seddik et al. (2020). An intuitive explanation of Assumption 1 is that the transformed random variable $f(\mathbf{x})$ for any $f : \mathbb{R}^d \to \mathbb{R}$ Lipschitz has a variance of order $\mathcal{O}(1)$. In particular, it implies that it does not depend on the initial dimension $d$. Furthermore, we place ourselves in the following large-dimensional regime.

**Assumption 2 (High-dimensional asymptotics)** *As $n \to \infty$, we consider the regime where $d = \mathcal{O}(n)$ and assume $d/n \to c_0 > 0$. Furthermore, we assume that $n_{tj}/n \to c_t^j$.*

Under Assumptions 1–2, Section 4 proposes a theoretical analysis of the classification error of SPCA-based continual learning algorithm.

## 4 THEORETICAL ANALYSIS AND OPTIMIZATION FRAMEWORK

### 4.1 THEORETICAL ANALYSIS OF THE SPCA-BASED CONTINUAL LEARNING

**Theorem 1** *Under Assumptions 1–2, for $\boldsymbol{x}$ a test data vector of class $j$ in task $\tau \in [t]$, following Assumption 1 with $\mathbb{E}[\boldsymbol{x}] = \boldsymbol{\mu}_\tau^j$ and $\boldsymbol{\Sigma}_{\boldsymbol{x}} = \boldsymbol{\Sigma}_\tau^j$,*

$$g(\boldsymbol{x}) - G_\tau^j \to 0, \quad G_\tau^j \sim \mathcal{N}(\mathfrak{m}_\tau^j, {\sigma_\tau^j}^2)$$

*in distribution where,*

$$\mathfrak{m}_\tau^j = \tilde{\boldsymbol{y}}^\mathsf{T} \mathcal{D}_{\boldsymbol{c}} \boldsymbol{M}^\mathsf{T} \boldsymbol{M} \boldsymbol{e}_{\tau j}^{[2t]}, \qquad {\sigma_\tau^j}^2 = \tilde{\boldsymbol{y}}^\mathsf{T} (\mathcal{D}_{\boldsymbol{s}} \mathcal{D}_{\boldsymbol{c}} + \mathcal{D}_{\boldsymbol{c}} \boldsymbol{M}^\mathsf{T} \boldsymbol{\Sigma}_{\boldsymbol{x}} \boldsymbol{M} \mathcal{D}_{\boldsymbol{c}}) \tilde{\boldsymbol{y}}$$

*where $\boldsymbol{e}_{\tau j}^{[2t]} = \boldsymbol{e}_{2 \times \tau + j}^{[2t]}$ is a one-hot vector, with the value at position $2 \times \tau + j$ is one. $\boldsymbol{s} = [s_1^1, \ldots, s_t^2] \in \mathbb{R}^{2t}$, with $s_\tau^j = \frac{1}{n} \mathrm{tr}(\boldsymbol{\Sigma}_\tau^j \boldsymbol{\Sigma}_{\boldsymbol{x}})$, $\boldsymbol{c} = [c_1^1, \ldots, c_t^2]$. $\mathcal{D}_{\boldsymbol{c}}$ and $\mathcal{D}_{\boldsymbol{s}}$ are the diagonal matrix with $\boldsymbol{c}$ and $\boldsymbol{s}$ as diagonal elements and $\boldsymbol{M} = [\boldsymbol{\mu}_1^1, \ldots, \boldsymbol{\mu}_t^2] \in \mathbb{R}^{d \times 2t}$.*

It is interesting to note that the theoretical high-dimensional performance of SPCA-based continual learning depends only on small data statistics. This is in line with many analyses generally conducted in the field of large-dimensional analysis. More specifically, in our case these sufficient statistics are: 1) the correlation between the task means through $M^\mathsf{T} M$, 2) the correlation between the covariance matrices through the vector $s$ and 3) the interaction between the means and covariance through $M^\mathsf{T} \Sigma_x M$. Note that these quantities incorporate the interaction between the tasks and the classes. We would like to mention that these statistics are easily estimated using empirical means and covariance matrices as explained in Remark 1 of the Appendix B. More importantly, performance depends on labels $\tilde{y}$ we have chosen to leave as changeable parameters. Next section will discuss the optimization of these labels in order to avoid *catastrophic forgetting* and resolve the *stability-plasticity* dilemma. Section 5 illustrates this experimentally.

Since $g(x)$ has a Gaussian limit centered around $\mathfrak{m}_\tau^j$, the asymptotic standard decision for $\mathbf{x}$ to be allocated to class 1 or class 2 for task $\tau$ is obtained by

$$g(\boldsymbol{x}) \underset{\mathcal{C}_2}{\overset{\mathcal{C}_1}{\lessgtr}} \gamma \tag{3}$$

for some threshold $\gamma$. To minimize the classification error, the optimal threshold $\gamma^\star$ for separating two Gaussians lies at the intersection between these two Gaussians and can be determined numerically as a function of $\mathfrak{m}_\tau^j$ and $\sigma_\tau^j$. The resulting classification error rate for task $\tau$ is given as

$$\epsilon_\tau(\tilde{\boldsymbol{y}}) = Q\left(\frac{\mathfrak{m}_\tau^1 - \gamma^\star}{\sigma_\tau^1}\right) + Q\left(\frac{\gamma^\star - \mathfrak{m}_\tau^2}{\sigma_\tau^2}\right) \tag{4}$$

with $\mathfrak{m}_\tau^j$ and $\sigma_\tau^j$ as defined in Theorem 1 and where $Q(z) = \frac{1}{\sqrt{2\pi}} \int_{-\infty}^z e^{-t^2/2} dt$ is the cumulative distribution function of standard normal distribution. As already mentioned, we defer the estimation of $\mathfrak{m}_\tau^j$ and $\sigma_\tau^j$ to Remark 1 in the Appendix B.

## 4.2 Optimization framework

Having access a priori to the theoretical classification error depending on the label $\tilde{\boldsymbol{y}}$, one can minimize the theoretical classification error as a function of $\tilde{\boldsymbol{y}}$. Interestingly, for the specific case where the weights of the tasks are $\boldsymbol{\omega} = e_\tau^{[t]}$ (which refers to optimizing the single task $\tau$), the optimal label has a closed form solution (see details of the derivation in Appendix B):

$$\tilde{\boldsymbol{y}}_\tau^{[t]\star} = \left(\mathcal{D}_s \mathcal{D}_c + \mathcal{D}_c M^\mathsf{T} \Sigma_x M \mathcal{D}_c\right)^{-1} \mathcal{D}_c M^\mathsf{T} M e_{\tau j}^{[2t]} \in \mathbb{R}^{2t}$$

where $\tilde{\boldsymbol{y}}_\tau^{[t]\star}$ denotes the optimal label for test task $\tau$ when $t$ tasks are seen.

**Theorem 2 ( On *catastrophic forgetting*)** *Under Assumptions 1–2, $\forall t$ and $\tau \in [t]$,*

$$\epsilon_\tau(\tilde{\boldsymbol{y}}_\tau^{[t+1]\star}) \leq \epsilon_\tau(\tilde{\boldsymbol{y}}_\tau^{[t]\star})$$

*where $\epsilon_\tau(\tilde{\boldsymbol{y}}_\tau^{[t]\star})$ is the classification error rate of task $\tau$ with optimal label after the model learned on task $t$.*

**Proof-sketch.** If we denote by $\bar{\tilde{\boldsymbol{y}}}_\tau^{[t+1]} = [\tilde{\boldsymbol{y}}_\tau^{[t]\star\mathsf{T}}, 0, 0]^\mathsf{T} \in \mathbb{R}^{2(t+1)}$, the vector $\tilde{\boldsymbol{y}}_\tau^{[t]\star}$ to which we add two-zeros for both classes of task $t + 1$, we have: $\epsilon_\tau(\tilde{\boldsymbol{y}}_\tau^{[t]\star}) = \epsilon_\tau(\bar{\tilde{\boldsymbol{y}}}_\tau^{[t+1]}) \geq \epsilon_\tau(\tilde{\boldsymbol{y}}_\tau^{[t+1]\star})$. The equality is due to the fact that the addition of label 0 to both classes of a new task doesn't contribute to the decision function (see equation 2). The inequality comes from that $\tilde{\boldsymbol{y}}_\tau^{[t+1]\star}$ is the optimal vector to minimize the classification error of task $\tau$ after seen task $t + 1$. Therefore, Theorem 2 ensures theoretically that catastrophic forgetting is prevented when adding more tasks. This will be verified empirically in Section 5.

In the case of a generic weight vector $\boldsymbol{\omega}$ (to optimize the multi-tasks), the optimization of $\tilde{\boldsymbol{y}}$ is performed using *Scipy* optimization toolbox (Bressert, 2012) with the cost function given as $\mathcal{L}(\tilde{\boldsymbol{y}}) = \sum_{\tau=1}^t \omega_\tau \epsilon_\tau(\tilde{\boldsymbol{y}})$, where the weights $\omega_\tau$ depending on the importance that user wants to put on task $\tau$ and $\epsilon_\tau(\tilde{\boldsymbol{y}})$ is given by equation 4. The optimization framework becomes flexible, allowing the *stability* and *plasticity* problems to be traded off and this will be illustrated in the Section 5.

### 4.3 ON THE MULTI-CLASS EXTENSION

Having an optimized framework for binary classifier, our approach to multi class setting aims to resort on *one-versus-one* or alternatively on *one-versus-all* approach. In a *one-versus-one* scheme, $\frac{1}{2}m(m-1)$ binary classifiers are trained (one for each pair of class $j$ and class $j'$, solving a binary classification problem). For each test sample, each binary classifier decides on or votes for the more relevant class. The test sample is then attributed to the class having the majority of votes. Each binary classifier is optimized by choosing appropriate labels and decision thresholds, thereby largely improving over the basal classifier performance. The *one-versus-all* approach applied to $m$ classes consists to build $m$ binary classifiers where for classifier $j \in [m]$, the first class is class $j$ and all other classes $j' \neq j$ are gathered into one common class. The final decision is obtained by taking the $argmin$ over the $m$ classifiers. An inherent problem of *one-versus-all* is the scales and alignments of the $m$ binary classifiers to be compared which are different. We solve this problem thanks to the theoretical statistics of $g(\boldsymbol{x})$ per classifier in addition to the optimization of the basal classifier. This procedure is thoroughly discussed in the Appendix C. Our algorithm (*one-versus-one*) is summarized as Algorithm 1 in Appendix and the code is available online [1].

## 5 APPLICATIONS

### 5.1 DATASETS DESCRIPTION

Through the experimental part, we will use 5 data sets (`Synthetic`, `Permuted MNIST` denoted *PMNIST*, `Split MNIST` denoted *SMNIST*, `Rotated MNIST` denoted *RMNIST*, `Split Fashion MNIST` denoted as *SFMNIST*). See more details in the Appendix D.

### 5.2 THEORETICAL VERSUS EMPIRICAL PREDICTION

The purpose of this section is twofold. Our first goal is to demonstrate the robustness of the theoretical analysis when applied to synthetic and real data by comparing the theoretical and empirical predictions. Secondly, we would like to highlight the significant improvement brought by the optimization of labels and illustrate that *catastrophic forgetting* is avoided. Figure 1 represents the empirical histogram of the decision function with the theoretical prediction for classical and optimized schemes. It is noticeable that the concentration random vector assumption is quite realistic since the theoretical and empirical predictions are close to each other.

Next we illustrate how *catastrophic forgetting* is avoided. To do so, we represent 3 examples in Figure 2. In the synthetic case we compare the average theoretical and empirical classification accuracy both for the classical and optimized scheme as a function of an increasing number of tasks. Here, optimization assumes equal weight for all tasks encountered. We can see a close fit between theory and practice while remarking that the performance of *OSCL* never decreases as opposed of its counterpart non optimized (SPCA) which forgets knowledge as new tasks are encountered. In the second and third picture (preformed on *PMNIST* and *RMNIST*), we represent the trajectory of the accuracy for each seen task as a function of the task number we observed so far. Table 1 shows the average forgetting measure where the values are all negative. We remark that by adding new tasks, we always improve the accuracy previously obtained, thus avoiding *catastrophic forgetting*.

Table 1: Average forgetting of each task on PermutedMNIST and RotatedMNIST (TIL $\downarrow$).

| Task | T1 | T2 | T3 | T4 | T5 | T6 | T7 | T8 | T9 | T10 |
|------|------|------|------|------|------|------|------|------|------|------|
| *PMNIST* | 0 | -1.019 | -0.915 | -0.473 | -0.493 | -0.150 | -0.272 | -0.206 | -0.149 | -0.221 |
| *RMNIST* | 0 | -1.63 | -1.840 | -0.660 | -0.480 | -0.444 | -0.330 | -0.110 | -0.219 | -0.129 |

### 5.3 ON THE FLEXIBILITY OF THE FRAMEWORK

In this section, we investigate the flexibility of the optimization framework by assigning different weights to different tasks. Considering a two-task binary classification problem, Figure 3 shows

---

[1]Code is released at https://github.com/huawei-noah/noah-research/tree/master/OSCL and https://gitee.com/mindspore/models/tree/master/research/AI-foundation/OSCL

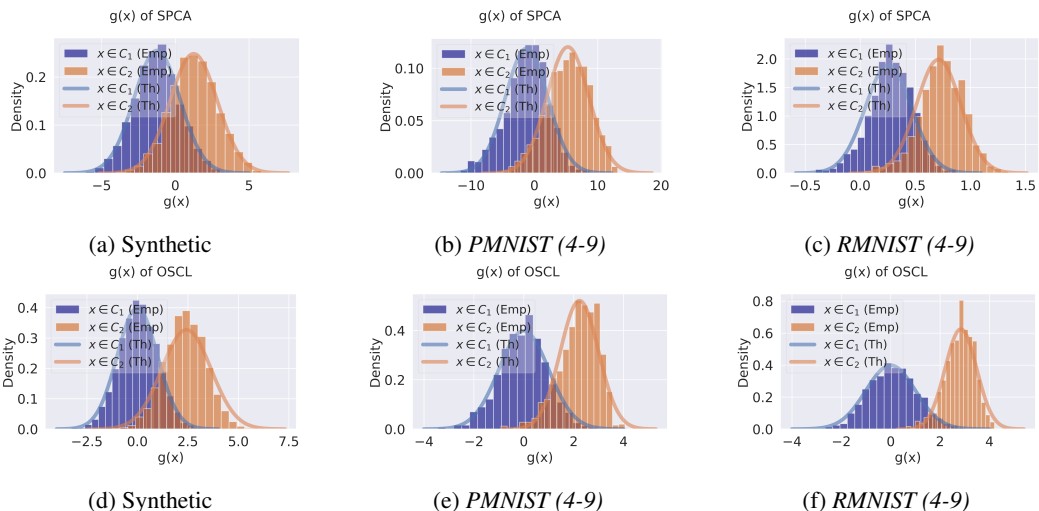

Figure 1: Empirical histogram versus theoretical prediction for the decision function $g(\boldsymbol{x})$ of task 2 in a two-task binary classification problem for the classical label (denoted SPCA) and for the optimized label (the optimization is performed with equal weight for both tasks). Close fit between theoretical and empirical predictions and better separation of the two classes with the optimized scheme in the bottom (denoted *OSCL*).

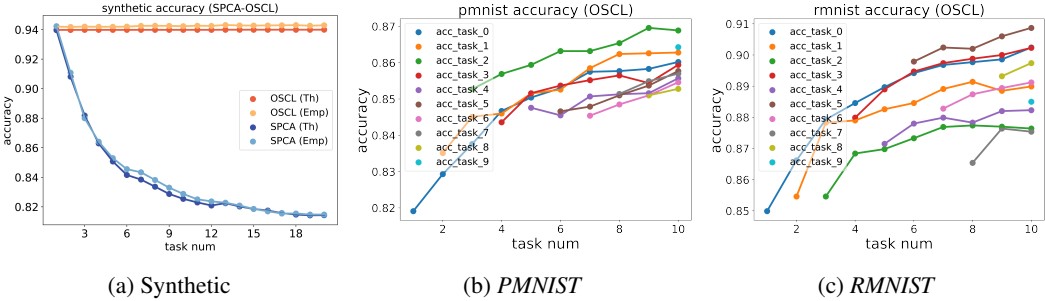

Figure 2: (Left) Empirical and theoretical classification accuracy of task 1 *w.r.t.* task number for the classical label (SPCA) and optimized label (*OSCL*). (Middle-Right) The accuracy trajectory of all seen tasks as a function of task number with the optimized label. In the learning process, the accuracy of tasks already seen increases over time.

the classification error of task 1 compared with task 2. Each coordinate of the point represents the classification error of task 1 and 2 along with the weight $(\omega_1, \omega_2)$ next to the point. We showed the performance of the weighting scheme with $(\omega_1, \omega_2) = \left( \frac{\alpha}{\sqrt{\alpha^2 + \beta^2}}, \frac{\beta}{\sqrt{\alpha^2 + \beta^2}} \right)$ and $\alpha$ goes from 0.8 to 0.2 and $\beta$ goes from 0.2 to 0.8 with a step size of 0.1 in Figure 3. In the same figure, a red cross represents the classification error of the classical label. A task that is given more weight is prioritized by the algorithm over a task that is given less weight. As a result of this flexibility, certain tasks can be favored at the user's discretion.

## 5.4 COMPARISON WITH SOME STATE OF THE ART ALGORITHMS

In this section, we compare our method to a number of state-of-the-art approaches to show the performance and the low computational cost of our algorithm. Four datasets and three scenarios (TIL, DIL and CIL) are used to fully validate the properties of our method.

For *SMNIST* and *PMNIST*, the experimental protocols following KJ & N Balasubramanian (2020), and the task number of *RMNIST* and *SFMNIST* are 10 and 5, respectively. More details about experimental setting will be discussed in Appendix D. From Table 2 and Table 3, as we can see, *OSCL*

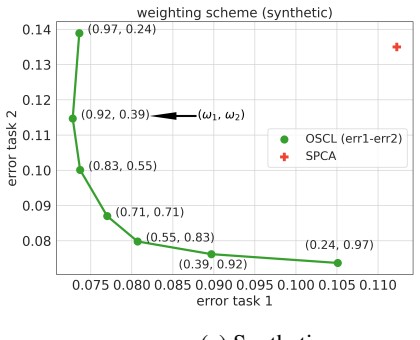

(a) Synthetic

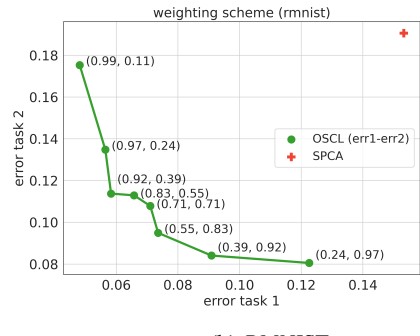

(b) *RMNIST*

Figure 3: Classification error of task 1 versus task 2. For *OSCL*, different weights are used to trade off the performance of tasks 1 and 2. Weights are formatted in $(\omega_1, \omega_2)$, with $\omega_1$ and $\omega_2$ representing the weight of task 1 and task 2, respectively. In general, the induced error is smaller when the task weight is greater.

Table 2: *Average accuracy* of the five baseline methods and the proposed one across 4 datatests.

| Dataset | *SMNIST* | | *PMNIST* | *RMNIST* | *SFMNIST* |
|---|---|---|---|---|---|
| scenario | CIL (↑) | TIL (↑) | DIL (↑) | DIL (↑) | CIL (↑) |
| EWC (Kirkpatrick et al., 2017) | $45.1 \pm 0.10$ | $98.64 \pm 0.22$ | $74.9 \pm 2.10$ | $78.35 \pm 0.73$ | $19.90 \pm 0.10$ |
| GEM (Lopez-Paz & Ranzato, 2017) | $86.7 \pm 1.5$ | $97.68 \pm 0.32$ | $82.5 \pm 4.90$ | **85.45±1.10** | $77.57 \pm 0.20$ |
| iCaRL (Rebuffi et al., 2017) | $89.9 \pm 0.90$ | $98.30 \pm 0.11$ | - | - | $80.70 \pm 1.29$ |
| SI (Zenke et al., 2017) | $45.2 \pm 0.15$ | $96.56 \pm 0.58$ | $74.5 \pm 2.50$ | $71.91 \pm 5.83$ | $20.00 \pm 1.50$ |
| MERLIN (KJ & N Balasubramanian, 2020) | $90.7 \pm 0.80$ | $97.4 \pm 0.30$ | **85.5±0.5** | - | - |
| *OSCL* (ours) | **91.2±0.10** | **98.0±0.43** | 77.41±0.81 | 72.7±0.42 | **90.0±0.22** |

Table 3: Running time on *SMNIST*.

| Methods | EWC | GEM | iCaRL | SI | MERLIN | *OSCL* |
|---|---|---|---|---|---|---|
| Running time (s) (↓) | 177.2±2.92 | 263.0±7.16 | 407.92±10.20 | 44.0±1.26 | 5218.96±1474.07 | 98.35±1.36 |

achieves competitive results with some baseline methods while being simpler in terms of complexity. Which means our method is more useful for low computational resource hardware devices. Since we use a linear classifier we have extracted HOG features (Dalal & Triggs, 2005) for some images to fairly compare with deep neural network based approaches. We also validate the effectiveness of using a deep neural network as a feature extractor and our method as a classifier. With the feature extracted by a pretrained convolutional neural networks, the performance will improve from 1.1% to 6.5% on different dataset compare with using HOG features, as detailed in the Appendix D.

## 6 CONCLUDING REMARKS

In this paper, a SPCA-based continual learning algorithm is analyzed from a high-dimensional perspective. Using the theoretical analysis, we propose an optimization of the labels that avoids *catastrophic forgetting*. In addition, the label optimization flexibly weight tasks during learning, thereby resolving the *stability-plasticity* dilemma. The method shows competitive results with the state of the art while being computationally inexpensive.

This work can be extended to the analysis of continual learning algorithm based on SVM and the integration of nonlinear kernels, as well as the combination with the neural networks.

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

## ABSTRACT

The appendix contains the main technical arguments omitted in the core of the article due to space limitations, and is organized as follows.

Section 2 complements the related works with a diagram illustrating some other CL methods. Section B derives the asymptotic classification error of the SPCA-based CL algorithm. To this end, Section B.1 proves the Gaussian distribution of the classification score under concentrated random vector assumptions. Section B.2 and B.3 computes respectively the first and second order moment of the classification score.

Section C complements the explanation of the multi-class extensions more precisely the *one-versus-all* approach. Section D provides additional experiments and additional insights into the experiments derived in the main paper.

## A  COMPLEMENTARY SECTION TO RELATED WORKS

To better summarize the section on the state of the art, we provide a rather didactic diagram in Figure 4 that we have constructed to classify the different methods of continual learning. Our method could be classified in the theoretical part but with important algorithmic implications of efficiency, flexibility and computation time.

## B  DERIVATION OF THE TECHNICAL RESULTS

In this section, we perform a theoretical analysis of SPCA-based continual learning. In order to derive the theoretical classification error, we need to understand the statistical behavior of the decision function $g(\boldsymbol{x}) = \boldsymbol{y}^\top \boldsymbol{X}^\top \boldsymbol{x}$ in particular its distribution and compute its first and second order statistics.

To that end we recall the assumption about the data (Assumption 3) and the growth rate of the dimension(Assumption 4).

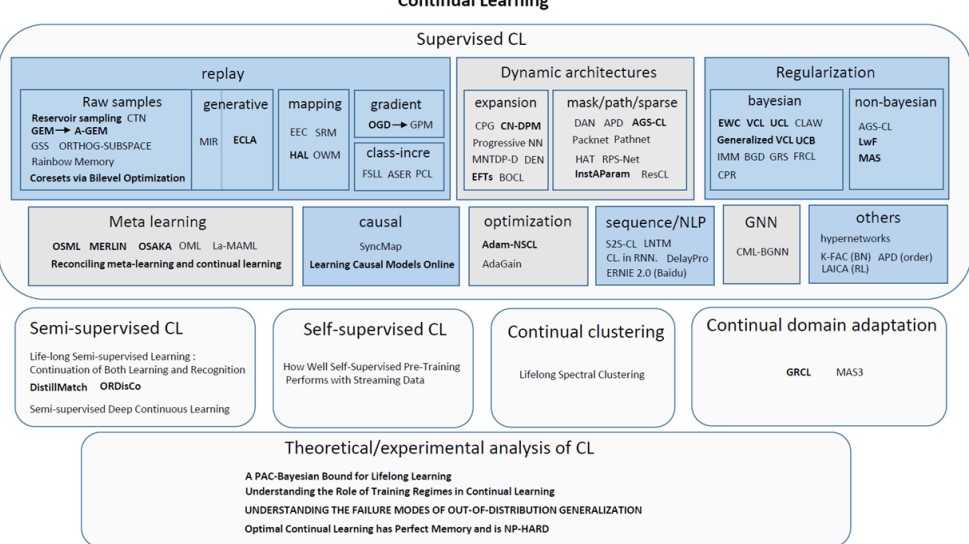

Figure 4: Summary of related works

**Assumption 3 (Concentration of $X$)** *For class $\mathcal{C}_j$, $j \in \{1, 2\}$ of task $t$, we assume that all vectors $\boldsymbol{x}_{t1}^j, \ldots, \boldsymbol{x}_{tn_{tj}}^j \in \mathcal{C}_j$ are i.i.d. such that $\mathrm{Cov}(\boldsymbol{x}_{t1}^j) = \boldsymbol{\Sigma}_t^j$ and $\mathbb{E}[\boldsymbol{x}_{t1}^j] = \boldsymbol{\mu}_t^j$. Moreover, we assume that there exist two constants $C, c > 0$ (independent of $n, d$) such that, for any 1-Lipschitz function $f : \mathbb{R}^d \to \mathbb{R}$,*

$$\mathbb{P}_{\boldsymbol{x} \sim \mathcal{D}(\boldsymbol{X})}\left(|f(\boldsymbol{x}) - m_{f(\boldsymbol{x})}| \geq t\right) \leq C e^{-(t/c)^2} \qquad\qquad \forall t > 0,$$

*where $m_Z$ is a median of the random variable $Z$.*

**Assumption 4 (High-dimensional asymptotics)** *As $n \to \infty$, we consider the regime where $d = \mathcal{O}(n)$ and assume $d/n \to c_0 > 0$. Furthermore, we assume that $n_{tj}/n \to c_t^j$ and we will denote $\boldsymbol{c} = [c_t^1, \ldots, c_t^2]$.*

### B.1 ON THE DISTRIBUTION OF THE DECISION SCORE.

In this section, the goal is to study the asymptotic behavior of $g(\boldsymbol{x})|\boldsymbol{x} \in \mathcal{C}_j$.

Since conditionally on the training data $\boldsymbol{X}$, $\boldsymbol{y}^\mathsf{T}\boldsymbol{X}^\mathsf{T}\mathbf{x}$ is expressed as the projection of the deterministic vector $\boldsymbol{X}\boldsymbol{y}$ on the Gaussian random vector $\mathbf{x}$, it follows that $\boldsymbol{y}^\mathsf{T}\boldsymbol{X}^\mathsf{T}\mathbf{x}$ is asymptotically Gaussian. The proof under the concentrated random vector unfolds from the following proposition.

**Theorem 3 (CLT for concentrated random vectors (Klartag, 2007; Fleury et al., 2007))** *If $\boldsymbol{x} \in \mathbb{R}^d$ is a concentrated random vector with $\mathbb{E}[\boldsymbol{x}] = \mathbf{0}$, $\mathbb{E}[\boldsymbol{x}\boldsymbol{x}^\mathsf{T}] = \boldsymbol{I}_d$ with an observable diameter of order $\mathcal{O}(1)$ and $\sigma$ be the uniform measure on the sphere $\mathcal{S}^{d-1} \subset \mathbb{R}^d$ of radius 1, then for any integer $k$, small compared to $d$, there exist two constants $C, c$ and a set $\Theta \subset (\mathcal{S}^{d-1})^k \subset \mathbb{R}^{d \times k}$ such that $\underbrace{\sigma \otimes \ldots \otimes \sigma}_{k}(\Theta) \geq 1 - \sqrt{d}C e^{-c\sqrt{d}}$ and for all $\boldsymbol{\theta} = (\boldsymbol{\theta}_1, \ldots, \boldsymbol{\theta}_k) \in \Theta$,*

$$\sup_{t \in \mathbb{R}} \left|\mathbb{P}(\boldsymbol{a}^\mathsf{T}\boldsymbol{\theta}^\mathsf{T}\boldsymbol{x} \geq t) - F_{0,1}(t)\right| \leq C d^{-\frac{1}{4}} \qquad \forall \boldsymbol{a} \in \mathbb{R}^k,$$

*where $F_{0,1}$ is the cumulative distribution function of the standard normal distribution $\mathcal{N}(0, 1)$.*

Since $g(\boldsymbol{x})$ is asymptotically Gaussian, we will focus on computing its first and second order moments.

## B.2 First order moment of the decision score.

Due to the independence between the training and test dataset $\boldsymbol{X}$ and $\boldsymbol{x}$, the mean for task $t$ reads as

$$
\begin{aligned}
\mathrm{m}_t^j &= \frac{1}{n}\mathbb{E}[g(\boldsymbol{x})|\boldsymbol{x}\in\mathcal{C}_j] \\
&= \frac{1}{n}\mathbb{E}[\boldsymbol{y}^\mathsf{T}\boldsymbol{X}^\mathsf{T}\boldsymbol{x}|\boldsymbol{x}\in\mathcal{C}_j] \\
&= \frac{1}{n}\boldsymbol{y}^\mathsf{T}\mathbb{E}[\boldsymbol{X}]^\mathsf{T}\mathbb{E}[\boldsymbol{x}|\boldsymbol{x}\in\mathcal{C}_j] \\
&= \frac{1}{n}\tilde{\boldsymbol{y}}^\mathsf{T}\boldsymbol{J}^\mathsf{T}\boldsymbol{J}\boldsymbol{M}^\mathsf{T}\boldsymbol{M}\boldsymbol{e}_{tj}^{[mT]} \\
&= \tilde{\boldsymbol{y}}^\mathsf{T}\mathcal{D}_c\boldsymbol{M}^\mathsf{T}\boldsymbol{M}\boldsymbol{e}_{tj}^{[2t]}
\end{aligned}
$$

where $\mathcal{D}_c$ stands for the diagonal matrix containing on its diagonal elements the elements of vector $\boldsymbol{c} = [n_{11}/n, \ldots, n_{2t}/n]$.

## B.3 Second order moment of the decision score.

The variance of the decision score $g(\boldsymbol{x})$ reads as

$$
\begin{aligned}
Var(g(\boldsymbol{x})) &= \mathbb{E}[g(\boldsymbol{x})^2] - \mathbb{E}[g(\boldsymbol{x})]^2 \\
&= \frac{1}{n^2}\mathbb{E}[\boldsymbol{y}^\mathsf{T}\boldsymbol{X}^\mathsf{T}\boldsymbol{x}\boldsymbol{x}^\mathsf{T}\boldsymbol{X}\boldsymbol{y}] - \frac{1}{n^2}\mathbb{E}[\boldsymbol{y}^\mathsf{T}\boldsymbol{X}^\mathsf{T}\boldsymbol{x}]\mathbb{E}[\boldsymbol{x}^\mathsf{T}\boldsymbol{X}\boldsymbol{y}] \\
&= \frac{1}{n^2}\mathbb{E}[\boldsymbol{y}^\mathsf{T}\boldsymbol{X}^\mathsf{T}\boldsymbol{C}_{\boldsymbol{x}}\boldsymbol{X}\boldsymbol{y}] - \frac{1}{n^2}\mathbb{E}[\boldsymbol{y}^\mathsf{T}\boldsymbol{X}^\mathsf{T}\boldsymbol{x}]\mathbb{E}[\boldsymbol{x}^\mathsf{T}\boldsymbol{X}\boldsymbol{y}]
\end{aligned}
$$

Moreover we have that

$$
\begin{cases}
\mathbb{E}[\boldsymbol{x}_i^\mathsf{T}\boldsymbol{C}_{\boldsymbol{x}}\boldsymbol{x}_j] = \mathbb{E}[\boldsymbol{x}_i]^\mathsf{T}\boldsymbol{C}_{\boldsymbol{x}}\mathbb{E}[\boldsymbol{x}_j] & i \neq j \\
\mathbb{E}[\boldsymbol{x}_i^\mathsf{T}\boldsymbol{C}_{\boldsymbol{x}}\boldsymbol{x}_i] = tr\left(\boldsymbol{C}_i\boldsymbol{C}_{\boldsymbol{x}}\right)
\end{cases}
$$

Therefore

$$
Var(g(\boldsymbol{x})) = \frac{1}{n^2}\mathbb{E}[\boldsymbol{y}^\mathsf{T}\mathcal{D}_T\boldsymbol{y}] + \frac{1}{n^2}\mathbb{E}[\boldsymbol{y}^\mathsf{T}\boldsymbol{J}\boldsymbol{M}^\mathsf{T}\boldsymbol{\Sigma}_{\boldsymbol{x}}\boldsymbol{M}\boldsymbol{J}^\mathsf{T}\boldsymbol{y}]
$$

where $\mathcal{D}_T = \begin{bmatrix} Tr(\boldsymbol{\Sigma}_{11}\boldsymbol{\Sigma}_{\boldsymbol{x}}) & 0 & \ldots & 0 \\ 0 & \ddots & \ldots & 0 \\ \vdots & \vdots & \ddots & \vdots \\ 0 & 0 & \ldots & Tr(\boldsymbol{\Sigma}_{mT}\boldsymbol{\Sigma}_{\boldsymbol{x}}) \end{bmatrix} \in \mathbb{R}^{n\times n}.$

We deduce the final expression of the variance as

$$
Var(g(\boldsymbol{x})) = \tilde{\boldsymbol{y}}^\mathsf{T}(\mathcal{D}_s\mathcal{D}_c + \mathcal{D}_c\boldsymbol{M}^\mathsf{T}\boldsymbol{\Sigma}_x\boldsymbol{M}\mathcal{D}_c)\tilde{\boldsymbol{y}}
$$

with $\boldsymbol{s} = [s_1^1, \ldots, s_t^2] \in \mathbb{R}^{2t}$, with $s_\tau^j = \frac{1}{n}\operatorname{tr}(\boldsymbol{\Sigma}_\tau^j\boldsymbol{\Sigma}_{\boldsymbol{x}})$

**Remark 1 (Estimation of $\mathrm{m}_\tau^j$ and $\sigma_\tau^j$)** *All the quantities defined in Theorem are known a priori except the scalar products $\boldsymbol{M}^\mathsf{T}\boldsymbol{M}$, $\boldsymbol{M}^\mathsf{T}\boldsymbol{\Sigma}_{\boldsymbol{x}}\boldsymbol{M}$ and the vector $\boldsymbol{s}$. Note that $\frac{1}{n}\operatorname{tr}(\hat{\boldsymbol{\Sigma}}_\tau^j\hat{\boldsymbol{\Sigma}}_{\boldsymbol{x}})$ is a consistent estimator for $s_\tau^j$ with $\hat{\boldsymbol{\Sigma}}_\tau^j$ and $\hat{\boldsymbol{\Sigma}}_{\boldsymbol{x}}$, the empirical covariance matrices. For large Gaussian data, $\boldsymbol{M}^\mathsf{T}\boldsymbol{M}$ can be effectively estimated as Tiomoko et al. (2020)*

$$
[\boldsymbol{M}^\mathsf{T}\boldsymbol{M}]_{qq} = \frac{4}{n_{\tau j}^2}\mathbf{1}_{n_{\tau j}}^\mathsf{T}\boldsymbol{X}_{[\tau]j;1}^\mathsf{T}\boldsymbol{X}_{[\tau]j;2}\mathbf{1}_{n_{\tau j}} + o_p(1)
$$

$$
[\boldsymbol{M}^\mathsf{T}\boldsymbol{M}]_{qq'} = \frac{1}{n_{\tau j}n_{\tau'j'}}\mathbf{1}_{n_{\tau j}}^\mathsf{T}\boldsymbol{X}_{[\tau]j}^\mathsf{T}\boldsymbol{X}_{[\tau']j'}\mathbf{1}_{n_{\tau'j'}} + o_p(1)
$$

*with $q = 2(\tau - 1) + j$ and $q' = 2(\tau' - 1) + j'$ different and $\boldsymbol{X}_{\tau j} = [\boldsymbol{X}_{\tau j;1}, \boldsymbol{X}_{\tau j;2}]$ an even-sized division of $\boldsymbol{X}_{\tau j}$.*

### B.4 DERIVATION OF THE OPTIMAL LABEL

In the case of identical covariance for both classes *i.e.,* $\boldsymbol{\Sigma}_\tau^1 = \boldsymbol{\Sigma}_\tau^2$, the optimal threshold $\gamma$ is given as

$$\gamma^\star = \frac{1}{2}(\mathrm{m}_\tau^1 + \mathrm{m}_\tau^2)$$

The associated classification error is given by

$$\epsilon_\tau = \mathcal{Q}\left(\frac{\mathrm{m}_\tau^1 - \mathrm{m}_\tau^2}{2\sigma_\tau}\right)$$

Therefore

$$
\begin{aligned}
\tilde{\boldsymbol{y}}_\tau^{[t]\star} &= \arg\min_{\tilde{\boldsymbol{y}}} \mathcal{Q}\left(\frac{\mathrm{m}_\tau^1 - \mathrm{m}_\tau^2}{2\sigma_\tau}\right) \\
&= \arg\max_{\tilde{\boldsymbol{y}}} \frac{\left(\mathrm{m}_\tau^1 - \mathrm{m}_\tau^2\right)^2}{2\sigma_\tau^2} \\
&= \arg\max_{\tilde{\boldsymbol{y}}} \frac{\tilde{\boldsymbol{y}}^\mathsf{T}\mathcal{D}_c\boldsymbol{M}^\mathsf{T}\boldsymbol{M}\boldsymbol{e}_{tj}^{[2t]}\boldsymbol{e}_{tj}^{[2t]\mathsf{T}}\boldsymbol{M}^\mathsf{T}\boldsymbol{M}\mathcal{D}_c\tilde{\boldsymbol{y}}}{2\tilde{\boldsymbol{y}}^\mathsf{T}(\mathcal{D}_s\mathcal{D}_c + \mathcal{D}_c\boldsymbol{M}^\mathsf{T}\boldsymbol{\Sigma}_x\boldsymbol{M}\mathcal{D}_c)\tilde{\boldsymbol{y}}}
\end{aligned}
$$

The previous optimization is known as the optimization of Rayleigh quotient (Li, 2015) which is convex and the optimal solution is given as

$$\tilde{\boldsymbol{y}}_\tau^{[t]\star} = \left(\mathcal{D}_s\mathcal{D}_c + \mathcal{D}_c\boldsymbol{M}^\mathsf{T}\boldsymbol{\Sigma}_x\boldsymbol{M}\mathcal{D}_c\right)^{-1}\mathcal{D}_c\boldsymbol{M}^\mathsf{T}\boldsymbol{M}\boldsymbol{e}_{tj}^{[2t]}$$

## C ON THE MULTI-CLASS EXTENSION

The literature Bishop & Nasrabadi (2006) describes broad groups of approaches to deal with classification with $m > 2$ classes. We focus here on the most common method, namely the one-versus-all approach. The complete optimization of one-versus-all being theoretically heavy to handle and demanding prior knowledge on the decision output statistics, the method inherently suffers from sometimes severe practical limitations; these are partly tackled here exploiting the large dimensional analysis performed in this article. In an *one-versus-one* setting, $\frac{1}{2}m(m-1)$ binary classifiers are trained (one for each pair $j, j'$ of classes, solving a binary classification). For each test sample, each binary classifier decides on – or "votes" for – the more relevant class. The test sample is then attributed to the class having the majority of votes. Although the number of binary classifiers is greater than in the one-versus-all approach, the training process for each classifier is faster since the training database is much smaller for each binary classifier. Besides, the method is more robust to class imbalances (since only pairwise comparisons are made) but suffers from an undecidability limitation in the case of equal numbers of majority votes for two or more classes.

In the one-versus-all method, focusing on Task $t$, $m$ individual binary classifiers, indexed by $\ell = 1, \dots, m$, are trained, each of them separating Class $\mathcal{C}_{t\ell}$ from the other $m - 1$ classes $\mathcal{C}_{t\ell'}, \ell' \neq \ell$. Each test sample is then allocated to the class index corresponding to the classifier reaching the highest of the $m$ classifier scores. Although quite used in practice, the approach first suffers a severe unbalanced data bias when using binary ($\pm 1$) labels as the set of negative labels in each binary classification is on average $m - 1$ times larger than the set of positive labels, and also suffers a center-scale issue when ultimately comparing the outputs of the $m$ decision functions, the average locations and ranges of which may greatly differ; these issues lead to undesirable effects, as reported in (Bishop & Nasrabadi, 2006, section 7.1.3)).

These problems are here simultaneously addressed: specifically, having access to the large dimensional statistics of the classification scores allows us to appropriately center and scale the scores. Each centered-scaled binary classifier is then further optimized by appropriately selecting the class labels (different from $\pm 1$) so to minimize the resulting classification error. See Figure 5 for a convenient illustration of the improvement induced by this centering-scaling and label optimization approach.

## C.1 ONE-VERSUS-ALL MULTI-CLASS OPTIMIZATION

For each target task $t$, in a one-to-all approach, $m$ SPCA-based CL binary classifications are solved with the target class $\mathcal{C}_{\tau\ell}^{\ell}$ (renamed "class $\mathcal{C}_{\tau1}^{\ell}$"), against all other $\mathcal{C}_{\tau2}^{\ell}$ classes (combined into a single "$\mathcal{C}_{\tau2}^{\ell}$ class"). Calling $g_{\mathbf{x}}^{(\ell)}$ the output of the classifier $\ell$ for a new datum $\mathbf{x}$, the class allocation decision is traditionally based on the largest among all scores $g_{\mathbf{x}}^{(1)}, \ldots, g_{\mathbf{x},t}^{(m)}$. However, this presumes that the distribution of the scores $g_{\mathbf{x}}^{(1)}$ when $\mathbf{x} \in \mathcal{C}_1$, $g_{\mathbf{x}}^{(2)}$ when $\mathbf{x} \in \mathcal{C}_2$, etc., more or less have the same statistical mean and variance. This is not the case in general, as depicted in the first column of Figure 5, where data from class $\mathcal{C}_1$ are more likely to be allocated to class $\mathcal{C}_3$ (compare the red curves).

By providing an accurate estimate of the distribution of the scores $g_{\mathbf{x}}^{(\ell)}$ for all $\ell$'s and all genuine classes of $\mathbf{x}$, Theorem 1 of the main article allows us to predict the various positions of the Gaussian curves in Figure 5. In particular, it is possible, for each binary classifier $\ell$ to center and scale $g_{\mathbf{x}}^{(\ell)}$ when $\mathbf{x} \in \mathcal{C}_\ell$. This operation averts the centering and scaling biases depicted in the first column of Figure 5: the result of the center-scale operation appears in the second column of Figure 5.

This first improvement step simplifies the algorithm which now boils down to determining the index of the largest $g_{\mathbf{x}}^{(\ell)} - m_{\tau1}^{(\ell)}, \ell \in \{1, \ldots, m\}$, while limiting the risks induced by the center-scale biases.

This being said, our theoretical analysis further allows to adapt the input labels $\tilde{y}^{(\ell)}$ in such a way to optimize the expected output. Ideally, assuming $\mathbf{x}$ genuinely belongs to class $\mathcal{C}_\ell$, one may aim to increase the distance between the output score $g_{\mathbf{x}}^{(\ell)}$ and the other output scores $g_{\mathbf{x}}^{(\ell')}$ for $\ell' \neq \ell$.

This is performed by maximizing the distance between the output score $g_{\mathbf{x}}^{(\ell)}$ for $\mathbf{x} \in \mathcal{C}_\ell$ and the scores $g_{\mathbf{x}}^{(\ell)}$ for $\mathbf{x} \notin \mathcal{C}_\ell$. By "mechanically" pushing away all wrong decisions, this ensures that, when $\mathbf{x} \in \mathcal{C}_\ell$, $g_{\mathbf{x}}^{(\ell)}$ is greater than $g_{\mathbf{x}}^{(\ell')}$ for $\ell' \neq \ell$. This is visually seen in the third column of Figure 5, where the distances between the rightmost Gaussians and the other two is increased when compared to the second column, and we retrieve the desired behavior. Algorithm 1 and C.1 summarizes the *OSCL* for *one-versus-one* and *one-versus-all* schemes

---

**Algorithm 1** Proposed *OSCL* algorithm (one-versus-one).

---

**Input:** Training samples of current task $t$ *i.e.,* $\boldsymbol{X}_t = [\boldsymbol{X}_t^1, \ldots, \boldsymbol{X}_t^m]$, $\boldsymbol{X}_\tau^\ell \in \mathbb{R}^{p \times n_{\tau\ell}}$, test data $\mathbf{x}$ and empirical means and covariance matrices of previous tasks $\hat{\boldsymbol{\mu}}_\tau^j, \hat{\boldsymbol{\Sigma}}_\tau^j$ for $\tau < t$.
**Output:** Estimated class $\hat{\ell} \in \{1, \ldots, m\}$ of $\mathbf{x}$.
**for** $j = 1$ **to** $m$ **do**
    **for** $j' \in \{1, \ldots, m\} \setminus \{j'\}$ **do**
        **Estimate:** $\boldsymbol{M}^\top \boldsymbol{M}$, $\boldsymbol{M}^\top \boldsymbol{\Sigma}_x \boldsymbol{M}$ and $\boldsymbol{s}$ according to Remark 1.
        **Create** optimal scores $\tilde{\boldsymbol{y}}^\star(j', j) = \arg\min_{\tilde{\boldsymbol{y}}} \sum_{s=1}^\tau \omega_s \epsilon_s(\tilde{\boldsymbol{y}})$.
        **Compute** classification scores according to equation 2, deduce the predicted class $c(j, j') = j$ or $c(j, j') = j'$ based on the decision rule in equation 3.
        **Update** the mean matrix $\boldsymbol{M}$ and $\boldsymbol{\Sigma}$ by sending the empirical means $\hat{\boldsymbol{\mu}}_t^j$ and covariance matrices $\hat{\boldsymbol{\Sigma}}_t^j$ for using in the next task
    **end for**
**end for**
**Output:** $\hat{j} = \displaystyle\operatorname*{mode}_{j', j \in \{1, \ldots, m\}} \{c(j, j')\}$.[2]

---

**Remark 2 (On the complexity of *OSCL*)** *The complexity of OSCL depends on the multi-class extension chosen. OSCL may be sequentially described as performing $\frac{1}{2}m(m-1)$ times (for the one-versus-one) and $m$ times (for the one-versus-all) the following procedure in 1) computing the decision score $g(\boldsymbol{x})$ as per equation 3, 2) estimation of $\boldsymbol{M}^\top \boldsymbol{M}$, $\boldsymbol{M}^\top \boldsymbol{\Sigma}_x \boldsymbol{M}$ and $\boldsymbol{s}$ , 3) optimization of $\tilde{\boldsymbol{y}}$. The complexity of computing $g(\boldsymbol{x})$ is a matrix-vector multiplication, yielding complexity $\mathcal{O}(n^2)$.*

---

[2]The mode of a set of indices is defined as the most frequent value. When multiple indices occur equally frequently, the smallest of those indices is considered by convention.

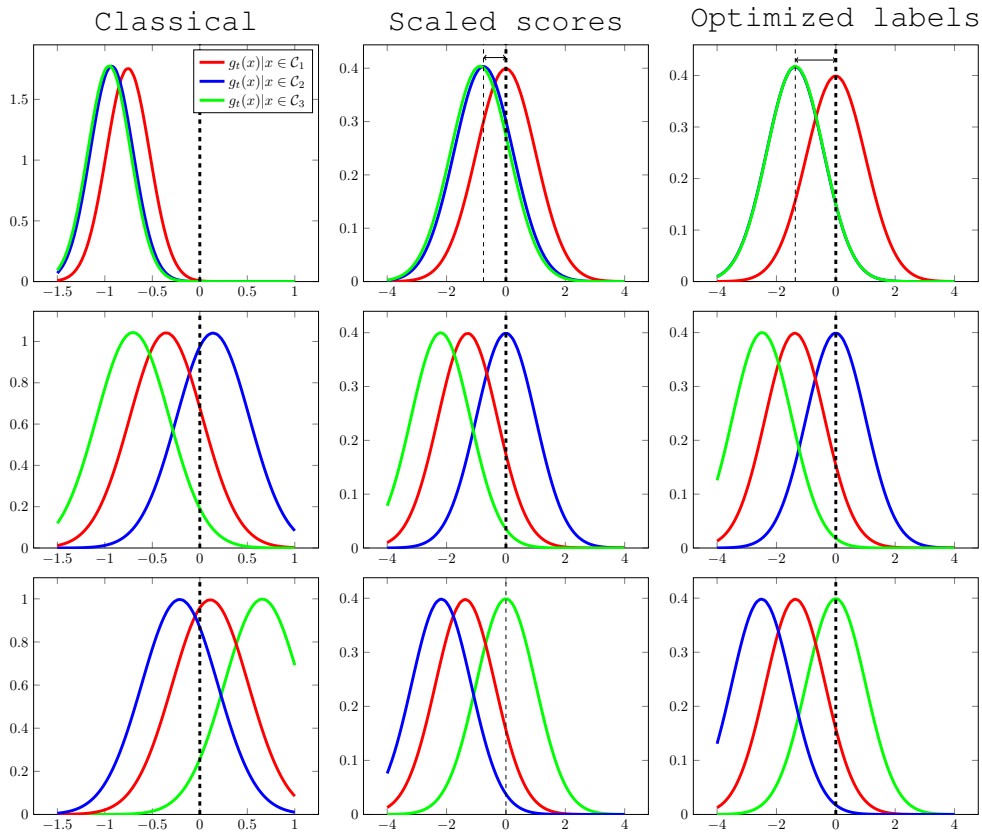

Figure 5: Test score distribution in a 2-task and 3 classes-per-task setting, using a one-versus-all multi-class classification. Every graph in row $\ell$ depicts the limiting distributions of $g_{\mathbf{x}}^{(\ell)}$ for $\mathbf{x}$ in different classes. Column 1 (Classical) is the standard implementation of the one-versus-all approach. Column 2 (Scaled scores) is the output for centered and scaled $g_{\mathbf{x}}^{(\ell)}$ for $\mathbf{x} \in \mathcal{C}_\ell$. Column 3 (Optimized labels) is the same as Column 2 but with optimized input scores (labels) $\tilde{y}^{\star(\ell)}$. Under "classical" approach, data from $\mathcal{C}_1$ (red curves) will often be misclassified as $\mathcal{C}_2$. With "optimized labels", the discrimination of scores for $\mathbf{x}$ in either class $\mathcal{C}_2$ or $\mathcal{C}_3$ is improved (blue curve in 2nd row further away from blue curve in 1st row; and similarly for green curve in 3rd versus 1st row).

---

**Algorithm 2** Proposed *OSCL* algorithm (one-versus-all).

---

**Input:** Training samples of current task $t$ *i.e.,* $\boldsymbol{X}_t = [\boldsymbol{X}_t^1, \ldots, \boldsymbol{X}_t^m]$, $\boldsymbol{X}_\tau^\ell \in \mathbb{R}^{p \times n_{\tau\ell}}$, test data $\mathbf{x}$ and empirical means and covariance matrices of previous tasks $\hat{\boldsymbol{\mu}}_\tau^j, \hat{\boldsymbol{\Sigma}}_\tau^j$ for $\tau < t$.

**Output:** Estimated class $\hat{\ell} \in \{1, \ldots, m\}$ of $\mathbf{x}$.

**for** $j = 1$ **to** $m$ **do**

    **Estimate:** $M^\top M$, $M^\top \Sigma_x M$ and $\boldsymbol{s}$ according to Remark 1.

    **Create** optimal scores $\tilde{\boldsymbol{y}}^\star(j) = \arg\min_{\tilde{\boldsymbol{y}}} \sum_{s=1}^t \omega_s \epsilon_s(\tilde{\boldsymbol{y}})$.

    **Compute** classification scores $g(\boldsymbol{x}, \ell)$ according to equation 2.

    **Compute** the theoretical statistics $\mathrm{m}_t^{j(\ell)}$ and $\sigma_t^{j(\ell)}$ as per Theorem 1, center and scale the decision function as $\frac{g(\boldsymbol{x}, \ell) - \mathrm{m}_t^{j(\ell)}}{\sigma_t^{j(\ell)}}$.

    **Update** the mean matrix $M$ and the covariance matrices $\Sigma$ by sending the empirical means $\hat{\boldsymbol{\mu}}_t^j$ and covariance matrices $\hat{\boldsymbol{\Sigma}}_t^j$ for use in the next task)

**end for**

**Output:** $\hat{\ell} = \arg\max_{\ell \in \{1, \ldots, m\}} g(\mathbf{x}; \ell)$.

---

*The computation of $\boldsymbol{M}^{\top}\boldsymbol{M}$ (as well as $\boldsymbol{M}^{\top}\boldsymbol{\Sigma}_{\boldsymbol{x}}\boldsymbol{M}$ and the vector $\boldsymbol{s}$) is of complexity $\mathcal{O}(dn+d)$ (estimation + product). The optimization of the labels being performed using the BFGS algorithm is of complexity $\mathcal{O}(t^3)$, $t$ being the number of tasks. The overall complexity is $\mathcal{O}(m^2 T^3) + \mathcal{O}(n^2 m^2)$ for the one-versus-one scheme and $\mathcal{O}(mT^3) + \mathcal{O}(n^2 m)$ for the one-versus-all. Note that competitors (using for most of them deep neural networks) can difficulty computes algorithmic complexity since it depends on the architecture chosen and depends on the internal optimization of the deep neural networks. But as confirmed with the running time (both for training and inference), our method achieves a lower computational cost.*

## D  EXPERIMENTAL PART

### D.1  THE DESCRIPTION OF DATA SET AND EXPERIMENTAL SET UP

The `Synthetic:` Gaussian mixture model with two classes with means $\mu_{\tau j} = (-1)^j \mu_\tau$ with $\mu_\tau = \beta_t \mu + \sqrt{1 - \beta_t^2} \mu^\perp$ for $\beta_t$ a relatedness parameter and with $\mu = e_1^{[p]}$, $\mu^\perp = e_p^{[p]}$.

The `Permuted MNIST` dataset has 70,000 images of handwritten digits from 0 to 9, with 60,000 being used for training and 10,000 for testing. Different from the original MNIST dataset, this dataset uses random permutations of pixels as inputs for each of the ten tasks.

The `Spilt MNIST` is based on 5 subsets of consecutive digits within the MNIST training data set. Learning to distinguish between two consecutive digits from 0 to 10 is the goal of the five tasks.

In the `Rotated MNIST`, the digits were rotated by an angle generated uniformly between 0 and 180 degrees and data for different tasks have different rotation angles. Based on this, the rotation angle and the factors of variation already included in MNIST are the factors of variation.

The `Fashion MNIST` dataset has the same size as MNIST, but is based on different (and more challenging) 10 classes. Similar to the `Split MNIST`, we will split the `Fashion MNIST` into 5 tasks, and the five binary classification tasks are: T-shirt/Trouser, Pullover/Dress, Coat/Sandals, Shirt/Sneaker, and Bag/Ankle boots.

Across all MNIST variants, 1000 samples were used as training data. For `Rotated MNIST` and `Permuted MNIST`, 10 tasks were generated. And regarding `Split MNIST` and *Split Fashion MNIST*, 5 tasks are generated, as in KJ & N Balasubramanian (2020); Sokar et al. (2020). Each experiment was run five times randomly to obtain the results. For a fair comparison, given that we are using a linear model, we used HOG (Dalal & Triggs, 2005) to extract features for `Rotated MNIST`, `Split MNIST`, and `Split Fashion MNIST`, and raw data for Permuted MNIST (without any feature extraction).

### D.2  EVALUATION METRICS

We evaluate our algorithm by *average accuracy $A$* across tasks. $A = \frac{1}{T}\sum_{t=1}^{T} A_t$, where $A_t = \frac{1}{t}\sum_{\tau}^{t} a_{t,\tau}$ and $a_{t,\tau}$ is the accuracy of task $\tau$ after the model is trained on $t^{th}$ task. We also use the metrics $F_t = \frac{1}{t-1}\sum_{j=1}^{t-1} (\max_{l \in \{1,\ldots,t-1\}} a_{l,j} - a_{t,j})$ to measure the average forgetting of our approach on task $t$ (Table 1) (KJ & N Balasubramanian, 2020).

### D.3  SUPPLEMENTARY EXPERIMENTS

Table 4: Accuracy of each task on SplitMNIST and Split FashionMNIST.

| Task | T1 | T2 | T3 | T4 | T5 |
|---|---|---|---|---|---|
| *SMNIST* | $98.34 \pm 0.73$ | $96.22 \pm 0.14$ | $92.86 \pm 0.05$ | $90.27 \pm 0.28$ | $85.99 \pm 0.12$ |
| *SFashionMNIST* | $99.39 \pm 0.20$ | $89.70 \pm 0.21$ | $89.12 \pm 0.13$ | $86.78 \pm 0.10$ | $85.02 \pm 0.22$ |

Table 5: Accuracy of each task on PermutedMNIST and RotatedMNIST.

| Task | T1 | T2 | T3 | T4 | T5 | T6 | T7 | T8 | T9 | T10 |
|---|---|---|---|---|---|---|---|---|---|---|
| *PMNIST* | $81.38 \pm 0.07$ | $80.57 \pm 0.14$ | $79.87 \pm 0.16$ | $78.91 \pm 0.24$ | $78.18 \pm 0.12$ | $77.05 \pm 0.21$ | $76.00 \pm 0.08$ | $75.12 \pm 0.11$ | $73.92 \pm 0.13$ | $73.18 \pm 0.09$ |
| *RMNIST* | $85.37 \pm 0.07$ | $84.04 \pm 0.22$ | $80.54 \pm 0.09$ | $78.17 \pm 0.14$ | $74.91 \pm 0.07$ | $71.36 \pm 0.11$ | $68.01 \pm 0.15$ | $62.60 \pm 0.21$ | $61.84 \pm 0.14$ | $60.87 \pm 0.18$ |

### D.3.1 COMBINATION WITH NEURAL NETWORK

We use an ordinary CNN model as pre-trained model, and it has five layers (2 convolution layers + 3 fully connected layers). The detail of model structure is as follows.

FashionMnistModel(

(layer1): Sequential(

  (0): Conv2d(1, 32, $kernel_{size}$=(5, 5), stride=(1, 1), padding=(1, 1))

  (1): BatchNorm2d(32, eps=1e-05, momentum=0.1, affine=True)

  (2): ReLU()

  (3): MaxPool2d($kernel_{size}$=2, stride=2, padding=0, dilation=1)

)

(layer2): Sequential(

  (0): Conv2d(32, 32, $kernel_{size}$=(3, 3), stride=(1, 1))

  (1): BatchNorm2d(32, eps=1e-05, momentum=0.1, affine=True)

  (2): ReLU()

  (3): MaxPool2d($kernel_{size}$=2, stride=2, padding=0, dilation=1)

)

(fc1): Linear(in=800, out=600, bias=True)

(drop): Dropout2d(p=0.25, inplace=False)

(fc2): Linear(in=600, out=120, bias=True)

(fc3): Linear(in=120, out=10, bias=True)

)

Table 6 summarizes the result obtained by using the aforementioned pretrained convolutional neural network together with the proposed method. The experimental settings and remarks are as follows:

Table 6: Average accuracy on three scenarios (TIL, DIL and CIL) with the HOG features and with the pretrained features.

| Task | *SMNIST* (TIL ↑) | *RMNIST* (DIL ↑) | *SMNIST* (CIL ↑) |
|---|---|---|---|
| HOG + optimized SPCA | $98.0 \pm 0.43$ | $72.7 \pm 0.42$ | $91.2 \pm 0.10$ |
| CNN feature + optimized SPCA | $\mathbf{99.1 \pm 0.40}$ | $\mathbf{79.2 \pm 0.11}$ | $\mathbf{95.3 \pm 0.04}$ |

- Considering we want to use the pre-trained model for MNIST data, we use an ordinary CNN model (2 convolution layers + 3 fully connected layers, more details about the model's structure see Appendix in main paper) and pre-train it on FashionMNIST data (60000 train data).

- FashionMNIST data is similar to the MNIST data. Both of them are in grayscale with the same shape. But the high-level semantic informations of these two datasets are totally different (handwritten digits vs clothing and personal adornment).

- We assume (or observe) that FashionMNIST and MNIST share similar semantic information at a low level. So an extracted feature is derived from the flattened output of the second convolution layer.

From Table 6, we can see a significant improvement after we use a neural network as a feature extractor. We believe that the improvement is caused by a better representation of the pre-train neural network. It is important to notice that extracted features must be chosen carefully, and several failures have been remarked in selecting high-level representations as features in our experiments.

