# OpenReview forum: "Optimizing Spca-based Continual Learning: A Theoretical Approach"
_ICLR.cc/2023/Conference — ICLR 2023 poster_

### Official Review · Reviewer_CZ28 · 2022-10-21

**Confidence:** 2
**Correctness:** 3
**Technical Novelty And Significance:** 3
**Empirical Novelty And Significance:** 3
**Recommendation:** 6

**Clarity, Quality, Novelty And Reproducibility:**

Poor clarity.
Good quality.
Nice originality of the work.

**Strength And Weaknesses:**


Strengths:

(a) This paper provides extensive theoretical analysis.

(b) The proposed method seems to be effective in the provided experiments.

Weaknesses:
I am not very good at supervised principal component analysis. Therefore, I talk about my feelings of this work.

(a) A more detailed description of supervised principal component analysis (SPCA) is necessary. What is SPCA? How is SPCA connected with CL?

(b) The description of proposed method is confusing. In the main file, no description of proposed method can be found, while they are placed in appendix. Clear description of proposed method is necessary.

(c) What does assumption 2 mean? What can we infer from it?

**Summary Of The Paper:**

This paper introduces a novel CL method based on Supervised Principal Component Analysis (SPCA). The main contributions can be fo summarized as:

(1) Proposing a simple continual learning algorithm, which is based on supervised principal component analysis.

(2) Providing a a theoretical analysis (exact classification error rather than bounds) using high dimensional statistics.

(3) Developing a develop a label optimization scheme.

(4) Several applications are presented

**Summary Of The Review:**

I am not very good at supervised principal component analysis. I recommend marginally above the acceptance threshold. I will change my score according to other reviewer's comments.

---

> ### Author Response · Authors · 2022-11-10
> **Answer to reviewer CZ28**
>
> We thank the reviewer for your interest in our ideas.
>
> **On details about SPCA and on connection with Continual Learning.**
>
> Originally, the SPCA was proposed as a supervised dimensionality reduction
> technique **(Barshan, Elnaz, et al., 2011)**. But it can also be seen as a supervised learning algorithm (like SVM, decision trees, and deep neural networks) whose decision function is given by Equation (1). It has been previously used in a single-task supervised setting.
>
> In this work, we provide an extension of SPCA to continual learning. The naive extension with a classical choice of labels leads to several problems while a theoretical analysis and an optimization of labels allow for preventing catastrophic forgetting, which is one of our major contributions. The advantage of SPCA in a Continual Learning context is its ease of transferring knowledge from previous tasks to the next in addition to its simplicity of theoretical analysis, which is a major advantage over more complicated algorithms
> such as SVM, LDA, logistic regression, or neural networks.
>
> **On the description of the method**
>
> We agree with the reviewer that the Algorithm describing the behavior of the proposed method should be included in the main paper (instead of the Appendix). In the updated version of the paper, we move the algorithm (Algorithm 1) which describes our procedure to the main article.
>
> **On the Assumption 2**
>
> Assumption 2 of the commensurable relationship between the number of samples and their dimension corresponds to a realistic regime and differs from classical asymptotic where the number of samples is often assumed to be exponentially larger than the feature size. Note that this chosen asymptotic regime classical in Random Matrix Theory fits most real-life applications and has been successfully applied in telecommunications (Couillet et al., 2011), finance (Potters et al., 2005), and more recently in machine learning **(Liao et al., 2019)**, **(Mai et al., 2020)**, **(Tiomoko et al., 2020)**. We have updated the paper to better explain this assumption.
>
> **Bibliography**
>
> - **(Barshan, Elnaz, et al., 2011)** Barshan, Elnaz, et al. "Supervised principal component analysis: Visualization, classification and regression on subspaces and submanifolds." Pattern Recognition 44.7 (2011): 1357-1371.
>  - **(Potters et al., 2005)** Bouchaud, Jean-Philippe, and Marc Potters. "Financial applications of random matrix theory: a short review." arXiv preprint arXiv:0910.1205 (2009).
>  - **(Couillet et al., 2011)** Couillet, Romain, and Florent Benaych-Georges. "Kernel spectral clustering of large dimensional data." Electronic Journal of Statistics 10.1 (2016): 1393-1454.
>  - **(Liao et al., 2019)** Louart, Cosme, Zhenyu Liao, and Romain Couillet. "A random matrix approach to neural networks." The Annals of Applied Probability 28.2 (2018): 1190-1248.
>  - **(Mai et al., 2020)** Mai, Xiaoyi, and Romain Couillet. "A random matrix analysis and improvement of semi-supervised learning for large dimensional data." The Journal of Machine Learning Research 19.1 (2018): 3074-3100.
>  - **(Tiomoko et al., 2020)** Tiomoko, Malik, Hafiz Tiomoko, and Romain Couillet. "Deciphering and optimizing multi-task learning: a random matrix approach." ICLR 2021-9th International Conference on Learning Representations. 2021.

---

> ### Author Response · Authors · 2022-11-19
> **Thanks to reviewer CZ28**
>
> Please accept our sincere thanks again for all your suggestions on our work. we hope our responses have answered your questions. We are happy to answer any questions you may have later.

---

### Official Review · Reviewer_w222 · 2022-10-23

**Confidence:** 5
**Correctness:** 2
**Technical Novelty And Significance:** 3
**Empirical Novelty And Significance:** 3
**Recommendation:** 8

**Clarity, Quality, Novelty And Reproducibility:**

The  theoretical result is promising, while many details need to clarify, and the writting needs to improve.

**Strength And Weaknesses:**

The theoretical result is sound and significant. While there exist some concerns as follows:
- Some mathematical notations seems inconsistent across the paper, making it hard to understand. E.g., the subscript order of c = [n11/n, …, n2t/n] in the appendix does not seem to be consistent with the previous description. Ds and Dc is not explained for readers.
- The formula of Q(z) is wrong. In the cumulative distribution function of standard normal distribution, The exponent of e should be -t^2/2. Whether this error will have an impact on subsequent theoretical proofs and experimental results.
- For the conclusion of Theorem 1, when the number of tasks t increases indefinitely, can the comparison of the information that SPCA-based methods and other continuous learning methods need to preserve be illustrated?
- Regarding the weight setting seems to be arbitrary, it is recommended to explain the rules for weight setting.
- The analysis of the experimental results is weak, e.g, in Table 1, why in DIL setting, OSCL substantially underperforms compared methods. In addition, since it is theoretically proven that the OSCL method avoids catastrophic forgetting, to illustrate this, it is recommended to include the average forgetting measure when presenting the results.
- Whether this method can be combined with a neural network?
- How to update the statistics in the online task environment？In my understanding, the dimension is increase when the number of tasks increase. How to use previous statistics to obtain current statistics?

**Summary Of The Paper:**

The paper propose a theoretical analysis of a SPCA-based continual learning algorithm using high dimensional statistics. Based on the theoretical analysis, they propose a label optimization process. Empirical evaluations highlight the effectiveness of the method.

**Summary Of The Review:**

Overall, this paper is interesting, since it provide a novel theoretical analysis of catastrophic forgetting. However, some important points (see above) need to clarify. The current version can not meet the requriement for publishing in ICLR.

---

> ### Author Response · Authors · 2022-11-10
> **Answer to reviewer w222**
>
> We thank the reviewer for the careful reading of the paper and for the interesting review.
>
> **On the mathematical notations errors.**
>
>  We thank the reviewer for pointing out some typos in the paper. Indeed, the correct order is $c = [n_{11}/n, ... , n_{t2}/n]$ instead of $c = [n_{11}/n, ... , n_{2t}/n]$. Similar $-t^2$ has been used instead of $-t^2/2$ in the expression of $Q(z)$. We would like to importantly insist on the fact that these typos don't have any implication on the theory. These have been updated in the new version of the paper. $\mathcal{D}_x$ has been defined in the notation section as the diagonal matrix with diagonal $x$ and $0$ elsewhere. We recall this definition in Theorem 1 in the updated version of the paper.
>
> **On the memory aspect when an increasing number of tasks.**
>
> The question raised by the reviewer is pertinent and interesting. For our method with $t$ tasks, we need to keep the statistics $\mathbf{M}$ and the covariance matrix $\mathbf{\Sigma}$. Even though this can be considered as big saving memory (especially for the covariance matrix), we see that using the identity covariance matrix, and scaling properly the data (by taking z-score normalization), the performance is almost unchanged for some datasets while saving huge memory size.
> For replay-based Continual Learning methods, in general, they need to keep samples from seen tasks which is also proportional to $t$.
> For Regularization-based and Dynamic architecture-based Continual Learning approaches, they may not be required to preserve the historical information, but either they will suffer catastrophic forgetting or the model size will increase with more tasks encountered.
>
> **On the weight setting.**
>
> In this context, the weight is the importance we assign to the corresponding task. Suppose we have two tasks and the first one is more critical, we can use the weights $\omega_1 > \omega_2$ to achieve this preference. In this sense, the weight can be given arbitrarily by the priority of the task which is set by the user. Thanks to that we can get the theoretical error of each task, based on which we can assign different weights to each task and then optimize a weighted objective function. To the best of our knowledge, we are the first work to achieve this characteristic.
>
> **On the experimental results is weak and average forgetting measure**
>
> Since our approach is a linear model and we only use HOG (Histograms of oriented gradients) to extract the feature, for particular data structures (images, text), the features extracted by a deep neural network can be the main asset for good performance. Therefore, it is understandable that our approach is not the best in some data sets. It should be especially emphasized that our approach is not designed to get the best results on all benchmarks, our core contributions are 1) proposing a method that can theoretically avoid catastrophic forgetting in task incremental learning scenarios; 2) providing flexible weight settings based on user preferences; 3) our approach is a lightweight approach for the end-side device (scenarios with more limited computational resources).
> We thank the reviewers to suggest we add the average forgetting measure in Table 1, and this has been performed in the updated version of the paper.
>
> **On the combined with a neural network.**
>
> Yes, our method can be combined with a neural network and this is the direction we plan to explore next step. A natural way is using a neural network as a feature extractor and our linear model as a classifier.
>
> **On update the statistics in an online environment.**
>
> For our method with $t$ tasks, we need to keep the statistics $\mathbf{M}$ and $\mathbf{\Sigma}$. To obtain current statistics, we just concatenate all the previous statistics with the new one of the new task then expand the dimension of $\mathbf{M}$ for example from $d\times 2t$ to $d\times 2(t+1)$.

---

> > ### Comment · Reviewer_w222 · 2022-11-13
> > **Thank you for your response**
> >
> > Thanks for your detailed response, and most of them address my concerns. While there still exist some concerns are not perfectly clarified. Firstly, the weight setting is vacuous, and the the rules for weight setting is not clear to readers. I suggest authers to present a practical algorithm to assign weights for each tasks. Seconly, I strongly suggest authers to use a neural network as a feature extractor and a linear model as a classifier. Current implementations ignore the task similarity by using HOG to extract feature. The catastrophic forgetting mainly  concentrates on neural network. Thirdly,  the update of statistics $\Sigma$ is not clear. Fourth, the bounds for online learning usually scale with $\mathcal{O}(\sqrt{T})$ or $\mathcal{O}(\log{T})$. Does it hold in this paper?

---

> > > ### Author Response · Authors · 2022-11-14
> > > **Answer to reviewer w222**
> > >
> > > Thank you for your reply again! And for your new questions, here are the answers.
> > >
> > > **on the rules for weighting setting**
> > > - The weight of the task is set according to the user's preferences. With Figure 3, the user can select a specific point (with the corresponding weights) based on their needs. A similar example is the P-R (Precision-Recall) curve, where one also chooses a suitable point on the P-R curve according to their requirement.
> > > - The value of the weights in Figure 3 is obtained by $(\omega_1, \omega_2) = (\frac{\alpha}{\sqrt{\alpha^2 + \beta^2}}, \frac{\beta}{\sqrt{\alpha^2 + \beta^2}})$ and $\alpha$ goes from 0.8 to 0.2 (0.8, 0.7, ..., 0.2) and $\beta$ goes from 0.2 to 0.8 (0.2, 0.3, ..., 0.8) with step size 0.1. We already update it in the main paper (Section 5.3). It should be noted that we have chosen these weights only as an example to show the relationship between the performances and weights. The most important information we want to show here is: the more weight we put on one task, the less error we will get for this task.
> > > - Because you have asked this question twice, if your question is not answered above, please let us know. We think there may be some misunderstanding in our expressions. If so, may you ask this question in a different way? Thanks
> > >
> > > **on the using neural network as a feature extractor**
> > >
> > > Firstly, we want to insist that we want to focus on the classifier in this paper which we can analyze theoretically by high dimensional statistics and have the guarantee for its performance. To the best of our knowledge, for a neural network, one cannot know its performance before testing it using data. But here for the SPCA-based continual learning method, we are able to know the theoretical error (the exact value rather than the bounds) even before testing it on data. Based on this good property, we proposed a label optimization scheme and have the theoretical guarantee of avoiding forgetting. However, if we use a neural network as a feature extractor here, it may break all the good properties of our method. So this is the major reason we don't want to mix them (the neural network as the feature extractor and the optimized SPCA as the classifier) together here.
> > >
> > > Secondly, we agree that neural networks suffer catastrophic forgetting in continual learning settings, but actually, it also happens for non-neural network approaches, like (SMO) self-organizing maps (Chen & Liu, 2018). In our paper, we found (in Figure 2 (a)) the catastrophic forgetting of the naive SPCA-based continual learning method. The way to avoid it is label optimization, which is one of the core contributions of this paper to the continual learning research community.
> > >
> > > Thirdly, there are three ways to combine with neural network and our method, and we will explore them in future works:
> > >
> > > - pre-trained neural network (e.g. pre-trained resnet18 or VGG16) as feature extractor + optimized SPCA as a classifier.
> > > - After seeing each task, retrain the model as the feature extractor + optimized SPCA as a classifier.
> > > - Updating a neural network with continual learning strategies (like EWC (Kirkpatrick, J. et al. 2017)) to avoid catastrophic forgetting as a feature extractor + optimized SPCA as a classifier.
> > >
> > > (Chen & Liu, 2018). Lifelong machine learning.Synthesis Lectures on Artificial Intelligence and Machine Learning, 12(3):1–207, 2018.
> > >
> > >  (Kirkpatrick, J. et al. 2017). Overcoming catastrophic forgetting in neural networks. Proceedings of the national academy of sciences, 114(13), 3521-3526.
> > >
> > > **on the updating of statistics $\Sigma$**
> > >
> > > Suppose we already seen $t$ tasks, the shape of $\Sigma_t$ is $d\times d\times 2t$. Where $d$ is the dimension of the feature, $t$ is the number of tasks. After that, the $t+1_{th}$ task will be encountered. To compute the new covariance matrix, we will concatenate the new statistics (which is in the shape of $d\times d\times 2$) from $t+1_{th}$ task to $\Sigma_t$. Therefore, the shape of $\Sigma_{t+1}$ will be $d\times d\times 2(t+1)$.
> > >
> > > **on the bounds of online learning**
> > >
> > > Since our algorithm is able to get the **exact classification error** (Equation 4 in the main paper) rather than the bounds, so we don't have a similar relationship of regret bounds with respect to the number of learning rounds $T$.
> > >
> > > Even though continual learning and online learning share some similarities (Both of them learn from continuous streams of data), they are different in the following ways ：
> > >
> > > - The optimization targets are different. In general, the goal of continual learning is to minimize empirical risk. But the goal of online learning is to minimize regret.
> > > - Continual learning tries to model the non-IID (not independent and identically distributed) data from multi-tasks, but online learning assumes an IID data sampling procedure and considers a single task.

---

> > > > ### Comment · Reviewer_w222 · 2022-11-16
> > > > **Thanks for your detailed clarification!**
> > > >
> > > > Thanks for your detailed clarification. It helps lot and addresses my concerns. I am happy to increase my score.
> > > >
> > > > I would like to see the improvement of combining your method with neural network. Besides, there exist many regret bounds for lifelong learning (not online learning), e.g., [1-3].
> > > >
> > > > [1] Alquier P, Pontil M. Regret bounds for lifelong learning. In AISTATS, 2017.
> > > > [2] Denevi G, Ciliberto C, Grazzi R, et al. Learning-to-learn stochastic gradient descent with biased regularization. In ICML, 2019.
> > > > [3] Denevi G, Stamos D, Ciliberto C, et al. Online-within-online meta-learning. In NeurIPS, 2019, 32.
> > > >
> > > > Reviewer w222

---

> > > > > ### Author Response · Authors · 2022-11-18
> > > > > **Answer to reviewer w222**
> > > > >
> > > > > We thank you for your reply and it's our pleasure to explain our work more clearly.
> > > > >
> > > > > **on the combination with neural network**
> > > > >
> > > > > We tried to combine a neural network and our method. And as we suggested last time, there are three possible ways to achieve this. We have chosen the first paradigm: pre-trained neural network as feature extractor + optimized SPCA as a classifier.
> > > > >
> > > > > The reasons we select this paradigm are as follows:
> > > > >
> > > > >
> > > > > - Pre-trained models are relatively common in the fields of computer vision (pre-trained ResNet (He, Kaiming, et al.), VGG (Simonyan, Karen, et al.) ) and natural language processing (BERT (Devlin, Jacob, et al.), GPT-3 (Brown, Tom, et al.) ), and these practices inspire us.
> > > > >
> > > > > - A fixed pre-trained model (no fine-tuning) can be considered as a common distribution transformation from $p(x)$ to $p(z)$, where $x$ is the raw feature and $z$ is the transformed feature. When combined with our approach, some of the good properties of our method can be fully preserved (e.g., avoiding catastrophic forgetting theoretically).
> > > > >
> > > > > The experimental settings and remarks are as follows:
> > > > >
> > > > > - Considering we want to use the pre-trained model for MNIST data, we use an ordinary CNN model (2 convolution layers + 3 fully connected layers, more details about the model's structure see Appendix in main paper) and pre-train it on FashionMNIST data (60000 train data).
> > > > > - FashionMNIST data is similar to the MNIST data. Both of them are in grayscale with the same shape. But the high-level semantic information of these two datasets is totally different (handwritten digits vs. clothing and personal adornment).
> > > > > - We assume (or observe) that FashionMNIST and MNIST share similar semantic information at a low level. So an extracted feature is derived from the flattened output of the second convolution layer.
> > > > >
> > > > >
> > > > > The experimental result:
> > > > >
> > > > > We test on different continual learning scenarios (Task incremental learning, Domain incremental learning, and Class incremental learning) and the results are as follows:
> > > > >
> > > > > |                            | SMNIST (TIL)     | RMNIST (DIL)    | SMNIST (CIL)   |
> > > > > |      ----                 |        ----                |    ----                   |  ----                   |
> > > > > |HOG + optimized SPCA        | 98.0 $\pm$ 0.43  | 72.7 $\pm$ 0.42 | 91.2 $\pm$ 0.10|
> > > > > |CNN feature + optimized SPCA| **99.1 $\pm$ 0.40**  | **79.2 $\pm$ 0.11**  | **95.3 $\pm$ 0.04** |
> > > > >
> > > > > From the table, we can see a significant improvement after we use a neural network as a feature extractor. And we believe that the improvement is caused by a better representation of the pre-train neural network. It is important to notice that extracted features must be chosen carefully, and we have had several failures in selecting high-level representations as features in our experiments. For reason of time, we didn't explore thoroughly about this direction (pre-trained neural network + optimized SPCA), and we believe that there will be a stronger feature extractor of the neural network which will bring us even better results.
> > > > >
> > > > > **On continual learning bounds**
> > > > >
> > > > > We thank the reviewer again for pointing out several interesting works on regret bounds derived for continual learning. We have added a section in the literature review to discuss these bounds. But as already briefly mentioned, the objective of our study is rather to have exact performances and no bounds. From this point of view, it is difficult to compare with this research axis since the tools and the aims are different.
> > > > >
> > > > > (He, Kaiming, et al.) "Deep residual learning for image recognition." Proceedings of the IEEE conference on computer vision and pattern recognition. 2016.
> > > > >
> > > > > (Simonyan, Karen, et al.) "Very deep convolutional networks for large-scale image recognition." arXiv preprint arXiv:1409.1556 (2014).
> > > > >
> > > > > (Devlin, Jacob, et al.) "Bert: Pre-training of deep bidirectional transformers for language understanding." arXiv preprint arXiv:1810.04805 (2018).
> > > > >
> > > > > (Brown, Tom, et al.) "Language models are few-shot learners." Advances in neural information processing systems 33 (2020): 1877-1901.

---

> > > > > ### Author Response · Authors · 2022-11-19
> > > > > **Thanks to reviewer w222**
> > > > >
> > > > > Please accept our sincere thanks again for all your suggestions on our work. We are not sure if the previous responses have answered your questions. We are happy to answer any questions you may have later.

---

### Official Review · Reviewer_6kJ9 · 2022-10-24

**Confidence:** 2
**Correctness:** 3
**Technical Novelty And Significance:** 3
**Empirical Novelty And Significance:** 3
**Recommendation:** 8

**Clarity, Quality, Novelty And Reproducibility:**

The paper lacks clarity, which cast a shadow on quality and novelty.

It would be better if the authors can explain how and why the theorem 1 and 2 can support the proposed method can prevent catastrophic forgetting.

**Details Of Ethics Concerns:**

None.

**Strength And Weaknesses:**

Strength:
1) using PCA and SPCA (or other dimensionality reduction methods) in continual learning can be beneficial, though it is not well motivated in this paper.
2) experimental results suggest performance improvement.

Weakness:
1) It is unclear why SPCA can prevent catastrophic forgetting in both the motivation and theoretical analysis.
2) There is a view (e.g. see https://arxiv.org/pdf/2011.05309.pdf) that SPCA can be simply replaced with a task loss plus the traditional PCA loss producing even superior result than Barshan’s method. Task loss is readily given in continual learning. Thus this may suggest a traditional PCA loss added to continual learning may do a better job that the proposed method.
3) Gradients Orthogonal Projection method is relevant to the proposed method, but is not analysis/compared sufficiently nor empirically.
4) I don't understand why labels need to be optimised. Labels are given by the tasks/datasets. The distribution of the labels are, in general, inaccessible, and the dataset is a sample of the distribution.





**Summary Of The Paper:**

The authors propose to use supervised PCA to tackle catastrophic forgetting in continual learning. They provide a theoretical analysis of the proposed method. They develop a label optimization scheme. They validate the method on four datasets and three scenarios (task incremental learning, domain incremental learning and class incremental learning).

**Summary Of The Review:**

I don't think the paper is ready to publish yet.

Post-rebuttal:
The original form of the paper was unclear and confusing. The authors answered all my questions well and these issues can be addressed in the camera-ready version. Hence I have increased my score.

---

> ### Author Response · Authors · 2022-11-10
> **Answer to reviewer 6kJ9**
>
> Thanks for the review. Below are some explanations about the remarks of the reviewer.
>
> **SPCA alone doesn't prevent catastrophic forgetting.**
>
> Supervised Principal Component Analysis (SPCA) is a supervised algorithm (like SVM, Logistic Regression, or Random Forest). It is possible to propose a natural extension of these algorithms to Continual Learning. But these naive extensions are generally prone to catastrophic forgetting. It is important to stress that SPCA is not the fundamental tool that prevents catastrophic forgetting but instead **its optimization**. One of our core contributions is that we propose an **optimized SPCA** with an optimal label that allows for preventing catastrophic forgetting.
>
> **Optimized labels in SPCA prevents catastrophic forgetting.**
>
> As correctly mentioned by the reviewer, in machine learning, labels are given (for example an image of a cat is given the label "cat" and an image of a dog the label "dog"). However one needs to choose a convention to encode the labels. Take a binary problem as an example and in continual learning setting one would naturally choose -1 for the first class and 1 for the second class of **all** tasks. But it is also allowed to choose the labels [-1, 1, -0.5, 0.5] (instead of [-1, 1, -1, 1]) for two tasks which will be equivalent to weight the second task by 0.5 and the first task by 1. Equation 4 of the main paper precisely says that the encoding $\tilde{y}$ is really important for the performance of continual learning. Intuitively, the 'label' can be seen as a weighting of all classes in all tasks. After optimization, classes that are closer to the categories in the test task will get more weight (the value of the label will be greater), and vice versa, they will get less weight (lower label value). But we can theoretically calculate how big these weights should be. Note that we are not the first in machine learning to propose label optimization (see **(tiomoko et al,. 2020)** for multi-task version) where the optimized label has been shown to avoid a negative transfer.
>
> **How do Theorem 1 and 2 support this affirmation?**
>
> Theorem 1 derives the mean $m_{\tau j}$ and the variance $\sigma_{\tau j}^2$ of the decision score of SPCA-based continual learning as a function of the encoding rule (or label) $\tilde{y}$, the data statistic $M$ and $\mathbf{\Sigma}_{ij}$.
> This allows deriving the theoretical classification error in equation 4 that precisely shows that the performance depends on $\tilde{y}$ on which we perform the optimization.
>
> Theorem 2 shows that if we choose the optimal label  $\tilde{y}^*$ given by closed form solution at the beginning of section 4.2, we ensure that the classification error of any task $\tau$ (when $t+1$ tasks are seen) is always lower than the classification of error of the same task $\tau$ (when $t$ tasks are seen), which is a rigorous proof of the capability to avoid catastrophic forgetting. This has been extensively shown in synthetic and real data in Section 5 and empirically illustrated by the average forgetting measure in Table 1 that we put in the new version of the paper.
>
> **On the Multiobjective SPCA.**
>
>  We would like a better clarification from the reviewer about the PCA-based method that would be better than the work performed here. We want to insist that we propose an approach based on large dimensional analysis that provides the optimal labels as a consequence. It will be ideal if the reviewer can be more precise on the alternative method that would be more relevant than the one proposed, or at least indications of why it would be better. In our case, we have shown that label optimization is optimal in the sense that it minimizes a convex cost function. Although we do not understand the method proposed by the reviewer we would like to have more insights into his method if it has been tried or if some theoretical guarantees have been derived. The referenced paper is about multiobjective SPCA and the indication of continual learning extensions is not mentioned nor similar theoretical derivations.
>
> **Gradient Orthogonal Projection is not relevant to our work.**
>
>  Let's recall that Gradient orthogonal projection takes gradient steps in the orthogonal direction when learning a new task to the gradient subspaces deemed important for the past tasks. We would like to mention that in our method we do not have any gradient computation or any constraints of orthogonality. In this sense gradient, orthogonal projection-based methods are completely different from the current work. However, we provide in the related works, a paragraph related to these methods.
>
> **Bibliography**
>  - **(tiomoko et al., 2020)** Tiomoko, Malik, Hafiz Tiomoko, and Romain Couillet. "Deciphering and optimizing multi-task learning: a random matrix approach." ICLR 2021-9th International Conference on Learning Representations. 2021.

---

> > ### Comment · Reviewer_6kJ9 · 2022-11-12
> > **Happy to increase the score**
> >
> > Thanks for the clarification. It helps a lot and I am happy to increase the score.
> >
> > Now I see the rationale of label optimisation and the theorems. Theorem 2 ensures catastrophic forgetting won't happen (since the error after seeing t+1 is always less or equal to the error after seeing t), and this is achieved by label optimisation (i.e. learning to reweigh the labels) with SPCA.
> >
> > Do you really mean "We will see that the classical choice of labels leads to catastrophic forgetting and does not allow us to solve the stability-plasticity dilemma" (page 5, just before eq(2))? This would invalidate all previous non-label optimisation methods.

---

> > > ### Author Response · Authors · 2022-11-12
> > > **Answer to reviewer 6kJ9**
> > >
> > > Thank you very much for your reply and we are very happy to explain our work more clearly.
> > >
> > > Regarding your question about our statement "classical choice of labels leads to catastrophic forgetting, ..." would invalidate all previous non-label optimization methods. The answers are:
> > >
> > > - What we want to convey here is: **for SPCA-based continual learning** with classical choice of labels will suffer catastrophic forgetting. This is supported by experimental results in Section 5 (Figure 2 (a)).
> > > - This statement of our claim is **only on SPCA-based continual learning**. For all previous non-label optimization methods, they have other way (to replay some samples, to limit the distance between old and new models, or to adapt the model structure to the tasks seen) to solve or reduce the catastrophic forgetting rather than label optimization. There is no conflict and our method is just another way to achieve the same goal (to avoid catastrophic forgetting)
> > > - To make things more clear, we already update our main paper, the updated statement is "We will see that the classical choice of labels **for SPCA-based continual learning** leads to catastrophic forgetting, ..."

---

> > > ### Author Response · Authors · 2022-11-19
> > > **Thanks to reviwer 6kJ9**
> > >
> > > Please accept our sincere thanks again for all your suggestions on our work. we hope our responses have answered your questions. We are happy to answer any questions you may have later.

---

### Official Review · Reviewer_62oS · 2022-11-04

**Confidence:** 3
**Correctness:** 3
**Technical Novelty And Significance:** 3
**Empirical Novelty And Significance:** 1
**Recommendation:** 6

**Clarity, Quality, Novelty And Reproducibility:**

Overall the paper is clearly written and the proposal is in high quality (though there is some concern in novelty). There are some suggestion to improve the clarity of the presentation (see my comments in weaknesses).

**Strength And Weaknesses:**

**Strengths**
- S1: (Even with strong assumptions) The theoretical analysis and a novel label optimization formulation seems novel.

**Weaknesses**
- W1: **Empirical validation is weak** -- Used benchmarks are all MNIST variants, thus in a small scale and relatively simple. Since the method is based on the strong assumption (See my summary), the empirical validation is quite weak.
- W2: **No discussion on the empirical comparison** -- Why the proposed method does not outperform the GEM in RMNIST.
- W3: **Presentation is somewhat unclear" -- Missing citation for the compared methods. Though the method of EWC, GEM, iCaRL and etc are quite well known, they should be properly referred either in the text or in the table.
- W4: **Recent replay-based methods (published in 2021-2022 -- see the list below) are largely ignored in review** - Some of them could be compared with but not all as they are evaluated in different set-ups.
  - Wang et al., "Continual Learning with Lifelong Vision Transformer", CVPR 2022
  - Hersche et al., "Constrained Few-shot Class-incremental Learning", CVPR 2022
  - Wan et al., "Continual Learning for Visual Search with Backward Consistent Feature Embedding", CVPR 2022
  - Tang et al., "Learning to Imagine: Diversify Memory for Incremental Learning using Unlabeled Data", CVPR 2022
  - Bang et al., "Rainbow memory: Continual learning with a memory of diverse samples", CVPR 2021

**Summary Of The Paper:**

The paper presents a theoretical analysis using SPCA, which predicts the performance of algorithm in advance. In addition, it propose a label optimization scheme to prevent the catastrophic forgetting with theoretical support (under the strong assumption of X are independent Gaussian random vectors with identity covariance).

**Summary Of The Review:**

Though the analysis and the proposed method is simple and limited in applicability, it presents an exact analysis on catastrophic forgetting and a novel label optimization formulation (though it's simple). The reviewer believe that this contribution is not trivial and meaningful to the community.

---

> ### Author Response · Authors · 2022-11-10
> **Answer to reviewer 62oS**
>
> We thank the reviewer for the interest in our work and the interesting feedback.
>
> **On the assumptions**
>
> Thanks for your summary of our work, but as already mentioned in the common answer to all reviewers, our assumption about the data matrix $\mathbf{X}$ is composed of concentrated random vectors with **general** covariance matrix, instead of independent Gaussian random vectors with identity covariance. Actually, our methods can work pretty well even on general covariance. More intuitions and explanations have been provided in the common answers to all reviewers about this assumption and are largely discussed in the new version of the paper.
>
> **On the weak empirical validation.**
>
>  Our model is just a linear model instead of a neural network. We think that using a linear model to handle **computer vision-oriented** benchmarks (cifar10, MiniImageNet, ...) is quite challenging and not the main goal of the paper. However, using a pre-trained deep neural network to extract features is a promising direction for handling such data.
>
>  **On the comparison with other methods.**
>
>  We would like to interestingly mention that our method being linear is not as efficient as end-to-end deep neural networks and can underperform sometimes on some specific datasets (structured data like images or text where deep neural networks extract good features). However, this happens only in scarce scenarios since in several datasets the method outperforms some state-of-the-art even being linear.
> Furthermore, our method enjoys several interesting properties that end-to-end deep neural network methods don't share.
>  - As shown in the running time comparison, the method achieves low computational cost. This is an important asset when deploying the model on low computational resource hardware devices.
>  - The theoretical analysis, insights with theoretical guarantees on catastrophic forgetting, and the flexibility of the method are some advantages of our method.
>
> As a side remark, we would like to insist on the fact that we do not want to achieve state-of-the-art results on all benchmarks but instead to show how and why a simple continual learning algorithm can prevent catastrophic forgetting theoretically with the adjustment of labels.
>
> **On related works.**
>
> We thank the reviewers for pointing out some details about the related works. In the updated version of the paper, we have clearly defined the different acronyms (EWC, GEM, iCaRL, ...) used in the table.
>
> **On replay-based methods.**
>
> We thank the reviewers for pointing out several references on replay-based methods. For conciseness and due to the large literature on continual learning, we select some state-of-the-art of each continual learning methodologies. But we definitively agree that the references mentioned by the reviewer could be an interesting addition to the related works. This has been performed in the updated version of the paper.
> We would like to mention that an important drawback of replay-based methods is that they need to keep some sample of already seen tasks which is a problem in terms of privacy protection and memory. Our algorithm doesn't need to store any previous examples. Besides, we want to stress that our method is generic and not designed for specific features (like images).

---

> > ### Comment · Reviewer_62oS · 2022-11-17
> > **Re: Answer to reviewer 62oS**
> >
> > Thank you for the detailed answers! My concerns are mostly addressed by the response but I still have a persisting concern that the limited applicability of the proposed method that is linear. Is the provided theoretical analysis possibly applicable to a non-linear model such as neural networks?

---

> > > ### Author Response · Authors · 2022-11-18
> > > **Answer to reviewer 62oS**
> > >
> > > We thank the reviewer for the reply, and we are happy to explain our work more clearly.
> > >
> > > **On the concern about limited applicability**
> > >
> > >
> > > - Firstly, we believe that the applicability really depends on which scenario one wants to use the model. For example, 1) on some end devices the computational resources are too limited to use complex models, and 2) real-time inference is required in some scenarios. For these real-world application scenarios, there are a lot of works on how to compress complex models. In this paper, we provide a **lightweight model with comparable performance**, which is actually a very competitive solution.
> > > - Secondly, we would like to clarify that end-to-end theoretical analysis of a deep neural network is not currently possible (to the best of our knowledge, there exists no work able to derive exact performance of deep neural networks a priori). In order to achieve a thousand miles, one must take the first step. Our work may serve as one of those steps. We will keep exploring this direction, and hopefully one day we will be able to analyze deep neural networks end-to-end. As such, our aim is to explain statistical phenomena in a simple model to better understand and inspire the study of more complex models.
> > > - Instead of end-to-end analysis of deep neural networks, another choice is to combine our method with a neural network (as feature extractor). A neural network can be used as a feature extractor and then put the extracted feature into our model. We tried it and here are the results.
> > >
> > > |                            | SMNIST (TIL)     | RMNIST (DIL)    | SMNIST (CIL)   |
> > > |  ----                      |       ----                 |   ----                   |  ----                    |
> > > |HOG + optimized SPCA        | 98.0 $\pm$ 0.43  | 72.7 $\pm$ 0.42 | 91.2 $\pm$ 0.10|
> > > |CNN feature + optimized SPCA| **99.1 $\pm$ 0.40**  | **79.2 $\pm$ 0.11**  | **95.3 $\pm$ 0.04** |
> > >
> > >
> > > The experimental settings and remarks are as follows:
> > >
> > > - Considering we want to use the pre-trained model for MNIST data, we use an ordinary CNN model (2 convolution layers + 3 fully connected layers, more details about the model's structure see Appendix in main paper) and pre-train it on FashionMNIST data (60000 train data).
> > > - FashionMNIST data is similar to the MNIST data. Both of them are in grayscale with the same shape. But the high-level semantic informations of these two datasets are totally different (handwritten digits vs clothing and personal adornment).
> > > - We assume (or observe) that FashionMNIST and MNIST share similar semantic information at a low level. So an extracted feature is derived from the flattened output of the second convolution layer.
> > >
> > > From the table, we can see a significant improvement after we use a neural network as a feature extractor. And we believe that the improvement is caused by a better representation of the pre-train neural network. It is important to notice that extracted features must be chosen carefully, and we have had several failures in selecting high-level representations as features in our experiments. For the reason of time, we didn't explore thoroughly about this direction (pre-trained neural network + optimized SPCA), and we believe that there will be a stronger feature extractor of the neural network which will bring us an even better result. But this reveals more engineering and experimentation than a well-motivated theoretical research approach.

---

> > > ### Author Response · Authors · 2022-11-19
> > > **Thanks to reviewer 62oS**
> > >
> > > Please accept our sincere thanks again for all your suggestions on our work. we hope our responses have answered your questions. We are happy to answer any questions you may have later.

---

### Author Response · Authors · 2022-11-10
**Answer to all reviewers**

We thank the reviewers for showing keen interest in our ideas, and for their thorough and quite valuable comments. In our post, we first provide a detailed point-by-point response to the comments common to all reviewers. Specific answers are then addressed to each reviewer in response to their remarks. We also upload the first version of the paper and supplementary material incorporating in blue some update.

**On the concentrated random vector assumption.**
We would like to interestingly recall that we base our theory on a concentration inequality (Assumption 1). This assumption is typical in most theoretical Random Matrix Theory works and includes many stronger assumptions such as Gaussian mixture. We would like to emphasize the fact that our assumption is much more general than Gaussian and encompasses a general covariance matrix (**beyond the identity covariance matrix!**).
An intuitive explanation of Assumption 1 is that the transformed random variables $f(\mathbf{x})$ for any $f: \mathbb{R}^{d} \rightarrow \mathbb{R}$ Lipschitz has a variance of order $\mathcal{O}(1)$ (**and not depending on the dimension $d$!**).
Although we are not aware of any formal method to check this assumption on real data,
a line of reasoning suggests that this concentration property is most likely present in many real data. Indeed, most machine learning algorithms are Lipschitz applications that transform data of high dimension $d$ into a scalar (the decision score).
If the data were not concentrated the decision score $f(x)$ would have a very large variance (depending on the dimension $d$) which would in turn lead to a random performance.
The fact that a machine learning algorithm is supposed to obtain non-trivial performance (different from randomness) combined with the fact that common machine learning algorithms are Lipschitz applications suggests that the concentration assumption is not meaningless for real applications.

Note that Generative Adversarial Network (GAN) images being concentrated by construction (being Lipschitz transforms of Gaussian data that are known to be concentrated) is the only known evidence for data that resemble real data as proven in **(Seddik et al., 2019)**.  We show in the experimental part the validity of our theory.
However, we make this assumption and a discussion clearer in the new version of the paper.

**Bibliography**
 - **(Seddik et al., 2020)** Random matrix theory proves that deep learning representations of GAN-data behave as gaussian mixtures
MEA Seddik, C Louart, M Tamaazousti, R Couillet
International Conference on Machine Learning, 8573-8582

---

### Decision · Program_Chairs · 2023-01-20

**Decision:**

Accept: poster

**Justification For Why Not Higher Score:**

Although it is great to see a theoretically grounded paper with good empirical support for the derived method, the fact that the results are presented on variants of MNIST and with linear methods only limits the audience for this paper.

**Justification For Why Not Lower Score:**

This paper is of value to the community, well rated by the reviewers and presents theoretically supported results.

**Metareview: Summary, Strengths And Weaknesses:**

This paper proposes a new continual learning extension over the supervised PCA method. This works by re-weighing the class labels for the different datasets in a way that prevents catastrophic forgetting. The re-weighing scheme is theoretically derived and experimentally tested.

It is great to see a paper that has strong theoretical grounding and is not purely empirical. Although the theory may not generalise to non-linear more complex methods, such as neural networks, the authors were able to show that the linear method can be successfully applied over features extracted by neural networks. Achieving good performance in the continual learning setting for sPCA is important, because this method can run on computationally constrained devices.

**Note From Pc:**

if the above contains the word "oral" or "spotlight" please see: "oral" presentation means -> notable-top-5% and "spotlight" means -> notable-top-25%. As stated in our emails, we are disassociating presentation type from AC recommendations